# Acetylcholine waves and dopamine release in the striatum

Lior Matityahu [1], Naomi Gilin [1], Gideon A. Sarpong[2], Yara Atamna [1], Lior Tiroshi [1], Nicolas X. Tritsch[3], Jeffery R. Wickens [2] & Joshua A. Goldberg [1] ✉

Striatal dopamine encodes reward, with recent work showing that dopamine release occurs in spatiotemporal waves. However, the mechanism of dopamine waves is unknown. Here we report that acetylcholine release in mouse striatum also exhibits wave activity, and that the spatial scale of striatal dopamine release is extended by nicotinic acetylcholine receptors. Based on these findings, and on our demonstration that single cholinergic interneurons can induce dopamine release, we hypothesized that the local reciprocal interaction between cholinergic interneurons and dopamine axons suffices to drive endogenous traveling waves. We show that the morphological and physiological properties of cholinergic interneuron – dopamine axon interactions can be modeled as a reaction-diffusion system that gives rise to traveling waves. Analytically-tractable versions of the model show that the structure and the nature of propagation of acetylcholine and dopamine traveling waves depend on their coupling, and that traveling waves can give rise to empirically observed correlations between these signals. Thus, our study provides evidence for striatal acetylcholine waves in vivo, and proposes a testable theoretical framework that predicts that the observed dopamine and acetylcholine waves are strongly coupled phenomena.

Striatal dopamine (DA) is essential for motivated behavior and reinforcement learning. Consistent with a role in these processes, the activity of DA neurons has been associated with reward prediction error[1,2] and other motivationally significant events[3]. Clinically, degeneration of DA neurons causes Parkinson's disease and striatal DA depletion causes motor symptoms[4]. To understand how DA contributes to these functions, the nature of DA neurotransmission has been extensively studied. The extensive arborization of DA axons[5], the high density of release sites[6], and the location of DA receptors and transporters some distance from release sites has led to the concept of DA as a volume-transmitted, global and spatially-uniform signal[7]. In contrast to the concept of a spatially-uniform signal, a recent imaging study has shown spatiotemporal traveling waves of DA[8]. These

traveling waves of DA concentration traverse the mediolateral (ML) aspect of the striatum, and are evident both in DA dynamics imaged using the genetically encoded DA sensor, dLight[9] and in DA axon activity, imaged using genetically encoded $Ca^{2+}$ indicators (GECIs). The functional significance of these waves is suggested by evidence that medial to lateral waves were associated with instrumental learning, while lateral to medial waves were associated with the reward delivery during classical conditioning[8]. How these waves arise and travel is currently unknown. Here we investigate the dynamical mechanism that gives rise to the formation and propagation of striatal traveling DA waves.

Studies indicating local modulation of DA release in the striatum provide an intriguing clue to the underlying mechanisms of striatal

[1]Department of Medical Neurobiology, Institute of Medical Research Israel – Canada, The Faculty of Medicine, The Hebrew University of Jerusalem, 9112102 Jerusalem, Israel. [2]Okinawa Institute of Science and Technology Graduate University, Okinawa, Japan. [3]Neuroscience Institute, New York University Grossman School of Medicine, New York, NY 10016, USA. ✉e-mail: joshua.goldberg2@mail.huji.ac.il

traveling DA waves. Several studies have shown gradual increases of DA concentration−DA ramps−during cued reward[10-15]. These gradual ramps differ from the transient, phasic increases in firing activity of DA cell bodies in the midbrain recorded during similar cued reward tasks[16] or locomotor acceleration[17]. Although the encoding function of DA ramps is hotly debated[18], there is mounting evidence that DA release is locally modulated, independent of DA cell firing at the soma[10-13,19-21].

Here we investigate the hypothesis that the traveling waves of DA are generated locally in the striatum. This hypothesis of a striatal origin of the DA waves is based on two observations. First, several studies have shown that striatal cholinergic interneurons (CINs) modulate the release of DA by actions at nicotinic acetylcholine (ACh) receptors (nAChRs) on DA terminals[22-25]. Striatal DA axons express α4β2 nAChRs, whose activation can "hijack" the axons and lead to local DA release[21-25]. These studies have shown that activation of striatal CINs causes striatal DA release in vitro, provided several CINs are activated synchronously. Thus, CINs might exert local control over DA release and DA-mediated behaviors[26,27].

Second, we previously found preliminary indications of wave-like dynamics−observed with GECIs−in the neuropil of striatal cholinergic interneurons (CINs) in freely moving mice[28,29]. The putative coexistence of ACh and DA waves in the striatum suggests a possible coupling of these two phenomena. Various studies have shown that CIN signaling is time-locked to DA signaling in the striatum, with some studies finding an out-of-phase or anti-phase relationship between the two signals[20,30-32]. The phase relationship may depend on whether, for example, a cue or reward is being presented[8,30]. The putative presence of coincident CIN and DA traveling waves will dictate a particular correlation structure between these two signals.

In the current study, we first present evidence for the existence of ACh waves in the striatum of mice. We then report that nAChRs extend the distance over which DA release can be detected after electrical stimulation by several hundred micrometers. Finally, we show, that activation of a single CIN suffices to induce local striatal DA release (i.e., synchrony among several CINs is not required). Based on these findings, we propose a dynamical scheme by which the local *reciprocal* interaction between CINs and DA fibers gives rise to traveling waves of both DA and ACh, creating temporal correlations that are similar to those observed empirically. We also discuss parameter regimes where this interaction between DA fibers and CINs can give rise to the formation of spatial Turing patterns[33] that manifest as "hills of activity" of DA and ACh, that may dynamically parcel the striatum into distinct functional regions of high vs. low concentrations of these two neuromodulators.

## Results

### Wave-like release of acetylcholine in the striatum

We have previously shown that activity within the cholinergic neuropil of the striatum, visualized microendoscopically with the GECI GCaMP6s in freely moving mice, is highly synchronized across the striatum and acts as a measure of collective CIN activity. At times we observed directional spreading of the GCaMP6s signal throughout the neuropil[28,29]. In light of the recent finding that DA release occurs in waves spreading along the ML axis of the striatum[8], we set out to determine whether striatal ACh release also forms waves. We conducted fluorescence imaging of a genetically encoded ACh sensor (GRAB-ACh3.0) expressed in the dorsal striatum (DS) of 3 head-fixed mice via a 3 mm diameter cranial window[34] (Fig. 1a, b). Visualization of the ACh signal demonstrated spatiotemporal patterns similar to those exhibited by DA in the DS[8], as the ACh signal could be visualized traveling across the DS (Supplementary Movie 1) primarily along the ML axis. In order to analyze the wave activity, we averaged the activity perpendicular to the ML axis (Fig. 1b), and tracked it over time. Diagonal streaks in the space-time rendition of this activity demonstrated the occurrence of waves that move along the ML axis (Fig. 1c, d).

Strikingly, tracking the location of the peak activity in space (Fig. 1c, d, dots) showed that location of the peak activity changed gradually in time with an instantaneous ML velocity that fluctuated in the ±10 mm/s range (Fig. 1d). Bootstrapping demonstrated that the motion of the peak activity of the ACh signal is inconsistent with random spatial activations[35] (Fig. 1e). We extracted wave events with a heuristic algorithm (see Methods), and estimated the distributions of wave durations and inter-wave intervals (Fig. 1f), which were consistent across mice (Supplementary Fig. 1a). From these distributions, we could estimate that waves occurred on average once every $5.2 \pm 0.5$ s (mean ± sem), and that their mean duration was $391 \pm 9$ ms. Importantly, the inter-wave intervals distributed across multiple time scales ranging from sub-second to 10 s of seconds, demonstrating that they occurred irregularly. Using the velocity curves (Fig. 1d, bottom), we extracted the distribution of the mean velocity of the waves, which were consistent across mice (Fig. 1g). Interestingly, approximately 80% of the waves spread from lateral-to-medial.

We obtained similar results by imaging another genetically-encoded ACh sensor (iAChSnFR) expressed in the DS via a cranial window with a 1 mm diameter field-of-view ($N = 2$ mice) or via a 1 mm diameter GRIN lens ($N = 4$ mice, Fig. 1h and Supplementary Movie 2). Here too, the ACh signal formed waves that exhibited a strong preference for travel in the lateral-to-medial direction (Fig. 1i) consistently across mice (Supplementary Fig. 1b, c). In these mice, ACh waves occurred on average every $8.6 \pm 1.2$ s, and their mean duration was $537 \pm 18$ ms (Fig. 1j and Supplementary Fig. 1d). The slightly lower velocities observed in these mice may result from differences in behavior or technique. First, the mice with iAChSnFR were imaged only while immobile, whereas those imaged with GRAB-ACh3.0 were allowed to run on a treadmill, which may be associated with faster waves. Second, because the former were imaged via a smaller (1 mm diameter) imaging aperture, the algorithm used to identify the waves may fail to identify waves whose spatial scale is larger than the aperture. Thus, the fact that both ACh and DA release exhibit wave activity in DS raises the possibility that these two signals are coupled, and may be generated by a joint mechanism, which is the central hypothesis of the current study.

### nAChRs control the spatial extent of striatal DA release

One way the spread of CIN and DA activity can share a common mechanism, is if CIN and DA activity are coupled, and that their coupling contributes to the spreading per se of the activity. While it is known that synchronous activation of CINs can activate nAChRs on DA axons to drive DA release[21-23,36], we wanted to determine whether this activation also extends the range of DA release. We therefore expressed a genetically-encoded DA sensor (GRAB-DA2m) in striatal DA axons, and measured the spatial extent of DA release in response to electrical stimulation of an acute striatal slice (Fig. 2a). The electrical stimulation triggered DA release several hundred micrometers from the bi-polar electrode (Fig. 2b, "control"). Estimation of the spatial profile of release, demonstrated that it fell off with a spatial scale of approximately 500 μm (Fig. 2c, "control"). Application of 10 μM of the nAChR antagonist mecamylamine (Fig. 2b, "mecamylamine") halved the spatial scale of DA release (Fig. 2c, "mecamylamine"), suggesting that the more distal release of DA depended on the recruitment of CINs in the vicinity of the stimulating electrode. To see if the activation of distant CINs is indeed responsible for causing DA release away from stimulating electrode, we next assessed how far CIN activation extends using transgenic mice expressing a GECI (GCaMP6f) in CINs. The spatial extent of the recruitment of CINs−estimated by measuring the spatial fall off of the GCaMP6f signal in response to the same stimulation parameters used to evoke DA release−was estimated at approximately 200 μm (Fig. 2c, "ChAT-GCaMP6f"). While differences in sensor properties could theoretically affect the comparison of spatial scales of GCaMP6f and GRAB-DA2m, the sensors differ

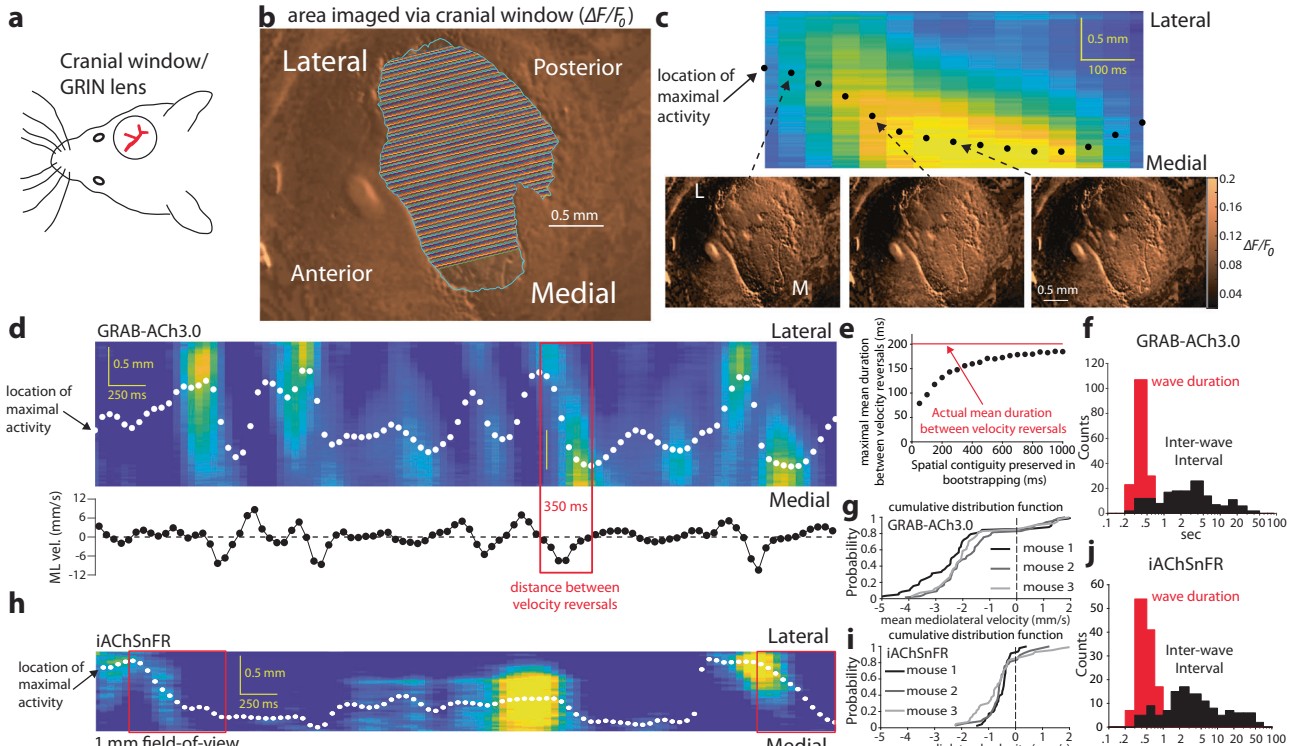

**Fig. 1 | Wave activity in striatal acetylcholine (ACh) release in head-fixed mice. a** Mouse with a cranial window or GRIN lens above dorsal striatum (DS). **b** Region-of-interest where the fluorescent ACh signals was averaged from pixels that ran perpendicular to the mediolateral (ML) axis (colored lines). **c** Top: space ($y$-axis)−time ($x$-axis) rendition of (z-scored) ACh release along the ML aspect display a gradual movement of activity from the lateral side to the medial side (dots indicate location of maximal activity along the ML aspect). Bottom: 3 frames of $\Delta F/F_O$ activity corresponding to 3 time points in the space−time rendition. **d** Top: 5-s-long space−time rendition of ACh release exhibit multiple diagonally oriented streaks representing wave-like progression of ACh release. White dots indicate the location of the peak activity. Bottom: spatial derivative of the location of peak activity provides an estimate of the instantaneous velocity of the ACh release along the ML aspect. Red rectangle indicates a time window when the ML velocity is negative, corresponding to a lateromedial wave. **e** Bootstrapping (see "Methods") resulted in maximal mean spurious durations between velocity reversals, that were well below the empirically observed mean duration between velocity reversals (red). **f** Distribution of duration of waves (red) and inter-wave intervals (black) pooled from the mice expressing GRAB-ACh3.0. **g** Cumulative distribution functions (CDFs) of the mean velocities of waves imaged in the mice expressing GRAB-ACh3.0. **h** 5 s-long space−time rendition of ACh release−imaged via a 1 mm diameter GRIN lens in a mouse whose DS expressed iAChSnFR−exhibits waves (red boxes). Space and time scales and white spots same as in **c**. **i** CDFs of the mean velocities of waves imaged in mice 1-3 expressing iAChSnFR. **j** Distribution of duration of waves (red) and inter-wave intervals (black) pooled from the mice expressing iAChSnFR. **g**, **i** (and Supplementary Fig. 1a, b) demonstrate that this experiment was repeated in 9 mice independently with similar results. Source data are provided as a Source data file.

primarily in their temporal properties[37–39], which should not strongly affect spatial decay. These findings show that activation of CINs in one region of the striatum promote the distant release of DA and suggest that local coupling between CINs and DA can cause the spread of DA release.

## Individual CINs can induce local DA release

The findings that ACh release exhibits wave-like properties similar to DA[8] and that CINs can spatially extend the release of striatal DA (Fig. 2) led us to hypothesize that local coupling between CIN and DA axons may underlie the wave-like activity. Moreover, we hypothesized that this coupling occurs throughout the densely intertwined arborization of CIN and DA axons, where localized release of ACh from an individual cholinergic axon may induce localized release of DA from nearby dopaminergic axons[21,25]. The prevailing view is that only synchronous activation of multiple CINs can induce localized DA release[21–23,25]. Indeed, we confirmed that synchronous optogenetic activation of CINs induced robust nAChR-dependent DA release (Supplementary Fig. 2a).

To test if activation of a single CIN can also induce DA release, we combined two-photon laser scanning microscopy (2PLSM) imaging of GRAB-DA2m, expressed selectively in dopaminergic axons, with patch-clamp recordings from individual CINs in acute striatal slices (Fig. 3a). We found that in 24% of the patched CINs ($n = 10/41$ CINs from $N = 10/$

18 mice), evoking a burst of action potentials (APs) caused a measurable increase in DA in the vicinity of the CIN. Measurements in various regions-of-interest within the axonal arbor showed DA release throughout the CIN's arbor (Fig. 3b), suggesting that a single CIN can influence DA release as far as its axonal field extends. In some cases, we were able to locate an individual stretch of DA axon that exhibits an even larger amplitude of release, presumably because the DA concentration is highest near the releasing axon that expressed the DA sensor (Fig. 3b, red region-of-interest). In 2 CINs, we were even able to observe DA release in response to a single AP (Fig. 3c). DA release could also be observed in 3 CINs in response to rebound spiking after hyperpolarizing the CIN, which may be important in the context of the pause response exhibited by CINs[30,40,41]. In 5 of the 10 CINs whose stimulation evoked DA release, the release occurred on multiple trials (that had to be separated by >1 min long intervals) within the nearby DA axonal arbor, although never at the same exact location. We confirmed in one of these CINs that DA release was blocked by the nAChR blocker mecamylamine (Fig. 3e), just as with the synchronous activation of CINs (Supplementary Fig. 2a).

To determine whether the frequent failures in DA release resulted from the unreliability of ACh release in response to activation of a single CIN, we repeated the above experiment but expressed GRAB-ACh3.0 in the dorsal striatum. We found that repeated activation of an

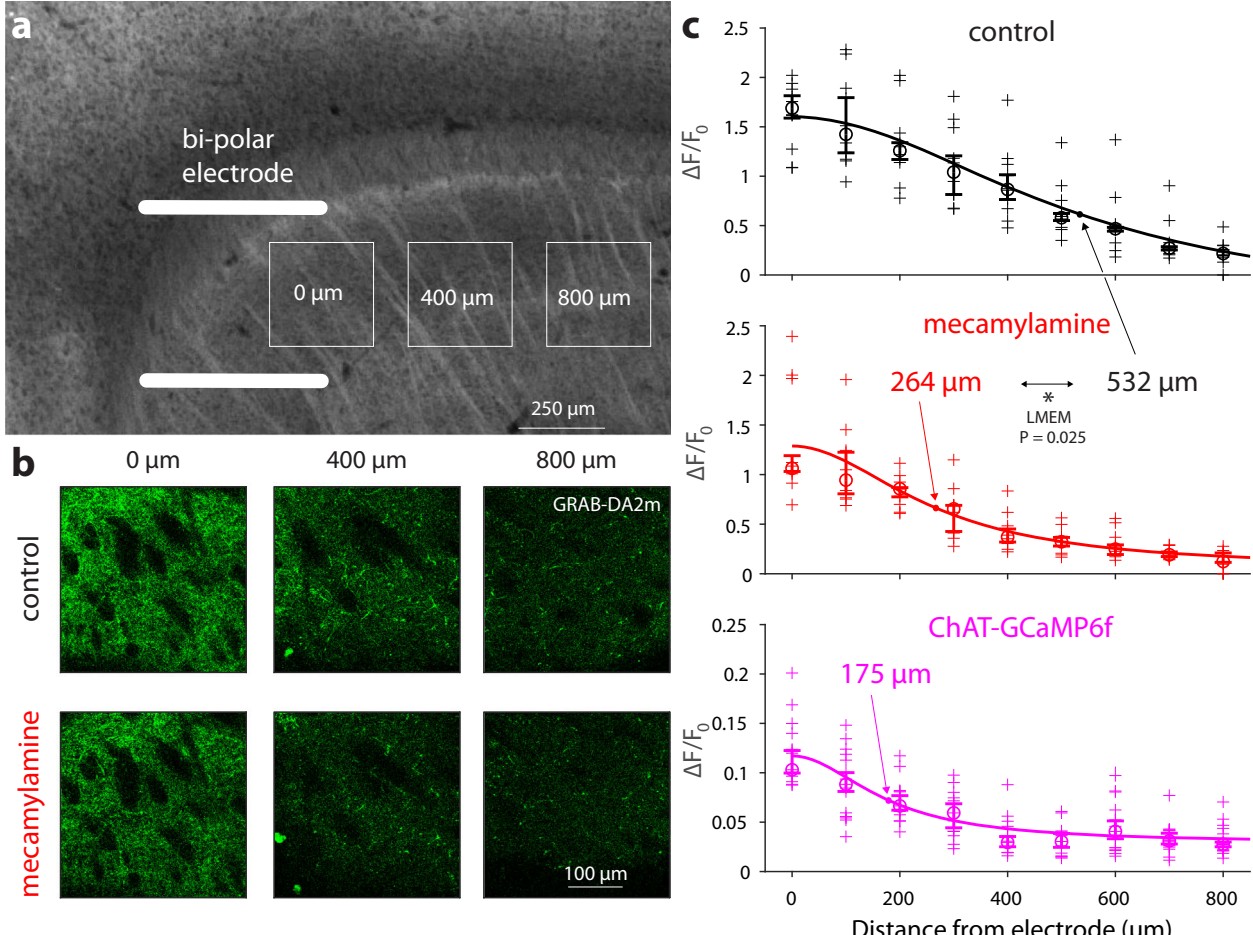

**Fig. 2 | nAChRs increase the spatial extent of striatal dopamine release.**
**a** Experimental design. Acute sagittal slice of striatum virally expressing GRAB-DA2m in DA axons is stimulated with a bi-polar electrode (drawn to scale). Maximal GRAB-DA2m fluorescence in response to electrical stimulation is measured within a 300 μm × 300 μm regions-of-interest, centered at 100 μm intervals from the electrode (0, 400, and 800 μm shown). **b** GRAB-DA2m fluorescence at various distances from the bi-polar electrodes in control (top) and after bathing slice in 10 μM mecamylamine, an nAChR antagonist (bottom). **c** Decay of GRAB-DA2m fluorescence ($\Delta F/F_O$) as a function of distance in control (top, $n = 11$ slices) and in mecamylamine (middle, LMEM, $t_{173} = 2.26$, $P = 0.025$), and decay of GCaMP6f fluorescence in cholinergic neuropil from ChAT-GCaMP6f mice as a function of distance (bottom, $n = 15$ slices). Crosses−individual measurements; circles−median; Confidence intervals indicate the $50\% \times (1 \pm 1/\sqrt{n})$ percentiles; solid curve−fit of Lorentzian to data points, from which the indicated spatial constants were extracted (see "Methods"). The GRAB-DA2m experiments were repeated independently in 2 mice, and the GCaMP6f experiment were repeated independently in 3 mice with similar results and were therefore pooled. Source data are provided as a Source data file. LMEM linear mixed-effects model.

individual CIN (every 5 s) reliably released ACh each time (Supplementary Fig. 2b). This suggests that the low repeatability of DA release that we observed is due to refractoriness of DA release sites after release, in line with previous studies. With repeated stimuli, DA release decreases sharply after the first stimulus[27,42–44] and stays decreased for up to 60 s[43]. Fluorescent false neurotransmitter experiments indicate that a single stimulus causes exocytosis of a large fraction of releasable vesicles (17%)[45] leading to a sharp decrease in DA release in response to subsequent stimuli[27,42,44]. Liu et al.[46] show that only the first action potential of a sequence triggers DA release. Moreover, the dopaminergic vesicle pool is slow to replenish (with a time constant of ~21 s)[47]. Finally, desensitization of nAChRs, which occurs more readily in acute striatal slices[27,44,48] may contribute as well.

Interestingly, we also observed spontaneous DA release events (e.g., Fig. 3e) that were not triggered within 100 ms of our stimulation, raising the possibility that the evoked responses were actually spontaneous ones that spuriously coincided with our stimulation. However, the observed rate of occurrence of DA release events is on average approximately 1 event per minute (i.e., a total of 87 spontaneous plus evoked events occurring during the cumulative 85.32 min of imaging, which). This rate is 30 times lower than reported in whole slice

imaging[21], but the area we imaged is typically two orders of magnitude smaller than the area imaged in that study, which can account for that discrepancy. With a rate of 1 event per minute, only 1.5 spontaneous events would be expected to occur within 100 ms of the 878 stimulation events delivered (i.e., 87 DA events × 878 stimulation events × 0.1 s coincidence window/5119.2 s), ruling out that the 26 evoked responses were spontaneous ones.

In summary, DA release events evoked by individual CINs in acute striatal slices are infrequent. Nevertheless, they are consistent with our working hypothesis that in the intact brain, where ACh and DA axons are not cut, and where nAChRs are not so desensitized[48] axo-axonal synapses between an individual CIN and its surrounding DA axon can cause local DA release.

## An extended model of local coupling between CINs and DA axons
Because the two neuromodulatory systems are made up of densely packed processes that interact locally throughout the striatum, we hypothesized that they behave like a nonlinear coupled reaction diffusion system that can give rise to traveling waves. We therefore constructed a reaction−diffusion model that replicates the main

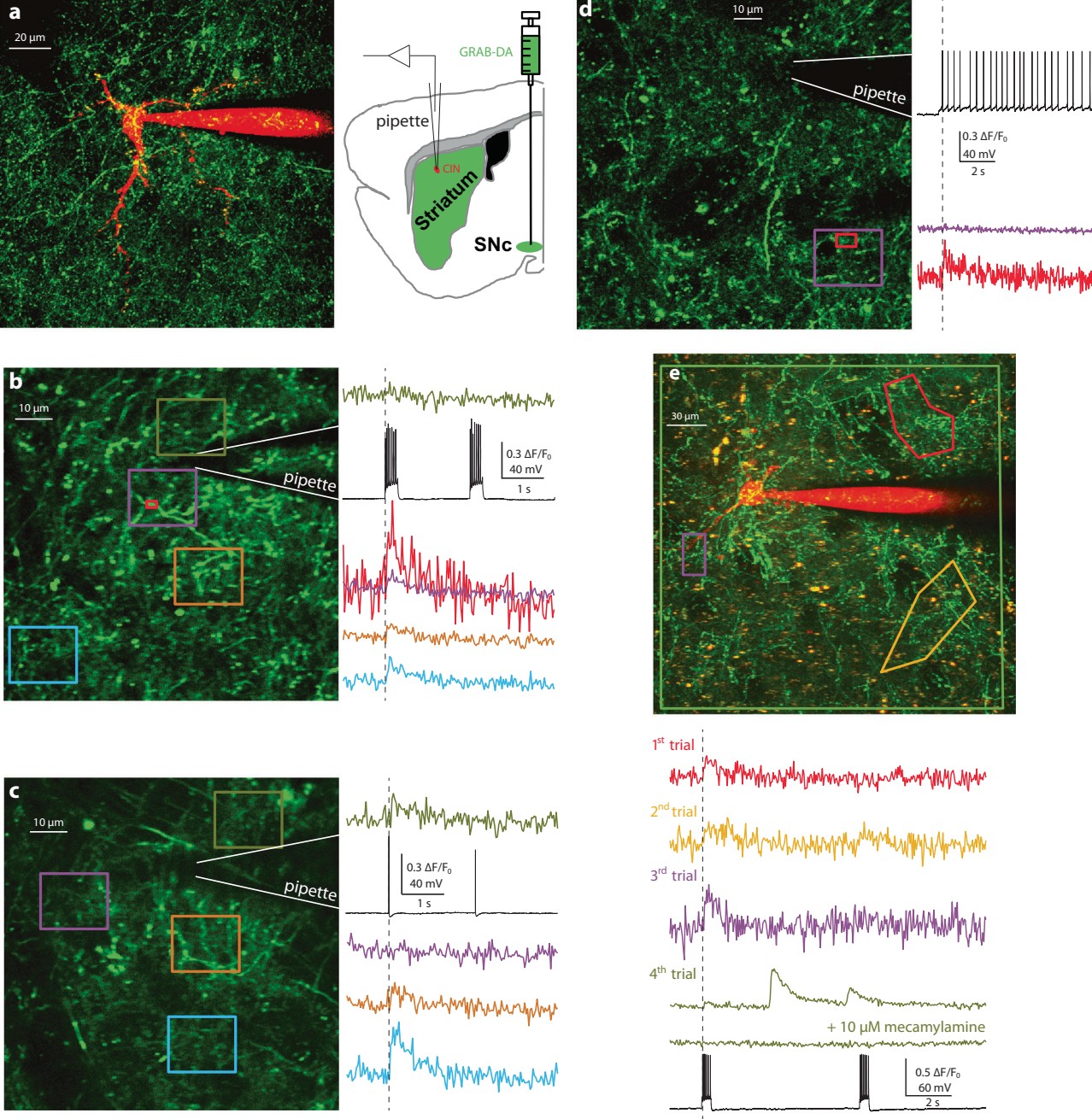

**Fig. 3 | Individual CINs can induce local DA release in acute striatal slices.**
**a** Experimental design. A CIN in an acute sagittal slice of striatum—visualized with 2PLSM—from a mouse whose substantia nigra pars compacta was inoculated with AAVs harboring GRAB-DA2m (green) is patched with a pipette containing Alexa-fluor 568 (red). **b** Depolarizing current pulses injected into a CIN hyperpolarized to quiescence elicited bursts of APs. Increases in DA can be observed in several (but not all) color-coded regions-of-interest that are 10 s of microns from the soma. A hot-spot of release can be observed in a small (red) region-of-interest around a patch of DA axon. **c** Even eliciting a single AP could trigger local DA release 10 s of microns from the soma. **d** DA release could be observed (in a very localized region-of-interest) also when a CIN was allowed to resume spiking after being hyperpolarized to quiescence. **e** DA release could be observed multiple times in the same slice but always at a different (color-coded) location, and was abolished by application of 10 µM mecamylamine (note the occasional spontaneous DA release events). DA release was always observed after the first discharge of CINs (vertical solid lines mark the precise time of the first action potential), but not after a second discharge when it occurred shortly after the first. The experiments were repeated in 18 mice with DA release successfully observed in 10 mice. Source data are provided as a Source data file.

features of the known coupling between CINs and DA axons, as described presently.

The dynamical scheme of our model is that of an activator inhibitor reaction diffusion (AIRD) system, which is known in chemistry and biology to give rise to traveling wave phenomena as well as to spatial Turing patterns[33,49]. As the name suggests, AIRD systems involve both activators and inhibitors. Because CINs activate nAChRs

on DA axons—even spontaneously under certain in vitro conditions[21,50–52]—we assume that CINs are the activators in the system. Conversely, the DA fibers are the inhibitors because DA release inhibits CIN activity via DA D2 receptors (D2Rs)[53,54].

Modeling CIN−DA axon interactions as a reaction diffusion system is justified by the dense, tortuous, space-filling nature of both DA and CIN axons, which have release sites every few microns[21,55]. This

**a**

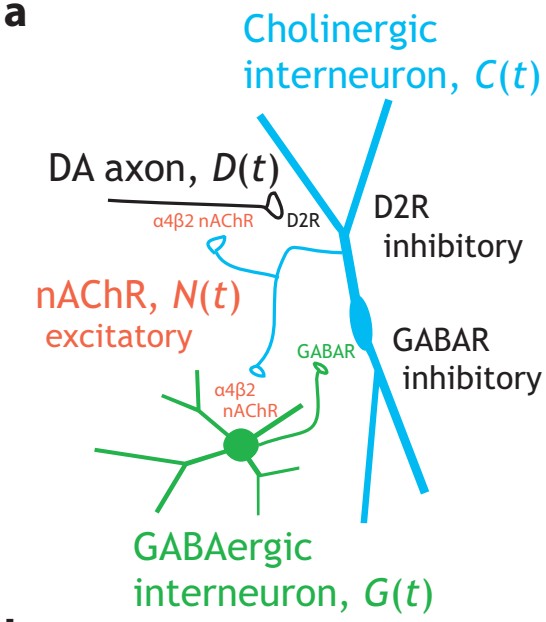

**b**

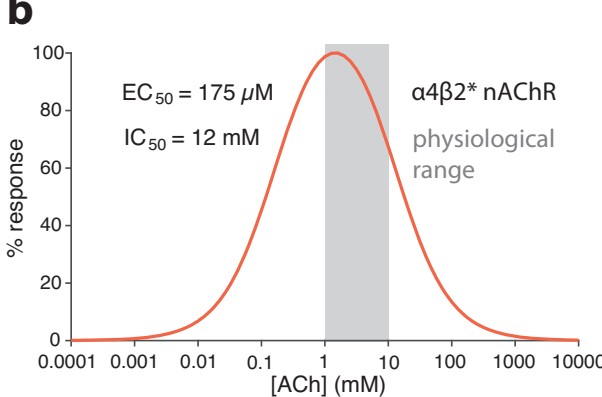

**c**

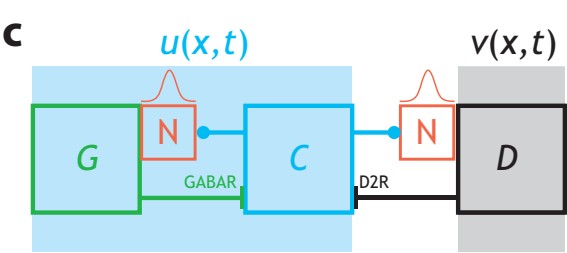

**Fig. 4 | Activator-inhibitor reaction diffusion (AIRD) model of local striatal interaction between CINs and DA axons. a** Circuit diagram of the reciprocal interaction between CINs and DA axons, as well as between CINs and GINs, and the dynamical variables use to denote the various elements, including the nAChRs. **b** Proposed inverted-U-shape dependence of nAChR activation on ACh concentration. The 1–10 μM range, indicated in gray, is hypothesized to be the physiological range, wherein an increase in concentration leads to a reduction in activation. **c** Extended AIRD model where the reciprocal interaction between CINs and GINs are subsumed into the variable $u(x,t)$, and the DA is represented by $v(x,t)$ (Eqs. 4–6). $v$ is governed by an inverted-U dependence on $u$, and $u$ is governed by an inverted-N dependence on itself.

structure combined with the fact that DA and ACh may diffuse some distance from release sites, lends support to our modeling them as continuous media (or syncytia) that fill space and interact within small, contiguous volumes. Finally, the activators in AIRD systems need to be "autocatalytic" (i.e., self-exciting through positive feedback). At first sight, this seems highly improbable, because CINs are known to be

mutually coupled by di-synaptic inhibition[56,57]. Nevertheless, we will propose one mechanism by which CINs may be self-exciting, due to the non-monotonic dependence of nAChRs on ACh concentration, and discuss other possibilities, as well.

### Overview of the model

The formalized model equations and justification of assumptions and parameter values is given below. An overview of the assumptions made in the construction of the model is as follows. To capture the activator-inhibitor relation we assume that CINs activate nAChRs on DA axons to increase DA release, and conversely that DA activates D2Rs on CINs to decrease ACh release. We assume that activation of nAChRs has an inverted-U shaped dependence on concentration of ACh with fast kinetics. To capture the reaction-diffusion interaction we assume that both DA and ACh axonal arbors can be represented by spatially extended variables, they fill the space and interact throughout the arbors, and the spread of activity is governed by the cable properties of the axons. The CINs are assumed to be self-exciting (or rather self-disinhibiting), by receiving inhibition from GABA interneurons (GINs) that flip the inverted-U shaped nonlinearity of nAChR activation into an inverted-N shaped dependence of CIN activation on ACh concentration.

### Derivation of the coupled CIN-DA model

We model CINs using a threshold linear rate model, with $C$ denoting their rate

$$\frac{dC}{dt} = -C + [I_C - \beta G - D]_+ \tag{1}$$

The autonomous firing of CINs[58] is realized by a constant input current, $I_C > 0$, and synaptic input to CINs can be modeled by temporal changes in $I_C$. $G$ represents the firing rate of the intrinsic source of striatal inhibition, which arises in vitro exclusively from GINs[59–61].

$\beta > 0$ represents the gain of the GIN-CIN connection, and $[x]_+ \stackrel{\text{def}}{=} \max(x,0)$. $D$ represents the DA axons that inhibit CINs via activation of DA D2Rs. The integration time constant of $C$ is assumed to be unity. The equation for $G$ is given by

$$\tau_G \frac{dG}{dt} = -G + [I_G + N]_+ \tag{2}$$

where $I_G > 0$, gives rise to the tonic activity of the GINs, such has been observed in low-threshold spiking interneurons (LTSIs)[61–63], but can also represent synaptic inputs to GINs. $N$ is the activation of the α4β2 nAChRs on the GINs. Note that here in Eq. 2 the threshold-linear gain function is superfluous and can be omitted. The integration time constant of the GINs is denoted by $\tau_G$. The equation for $N$ is given by

$$\tau_N \frac{dN}{dt} = -N + h(C) \tag{3}$$

where the integration time constant of the nAChRs is denoted by $\tau_N$ (Fig. 4a). The function $h(\cdot)$ represents the dependence of the α4β2 nAChRs activation on the activation of CINs. α4β2 nAChRs activation has an inverted-U shaped dependence on to concentration of ACh, and behaves roughly like $n([ACh]) = \{(1 + \frac{EC_{50}}{[ACh]})(1 + \frac{[ACh]}{IC_{50}})\}^{-1}$ with $EC_{50} = 175$ μM and $IC_{50} = 12$ mM[64,65]. Note that $n(\cdot)$ peaks in the 1 mM range. ACh is thought to reach concentrations of 1–10 mM at the post-synaptic end of the synaptic cleft of the neuromuscular junction[66] and similar concentrations in the proximity of CIN release sites in the striatum[55]. Because of the wide range of concentrations, the inverted-U region of this function—namely, where an increase in ACh concentration can lead to *less* activation of nAChRs—is physiologically relevant

(Fig. 4b). Assuming that there is a monotonic (increasing) dependence of striatal ACh concentration on the activity of CINs, we will model $h(\cdot)$ as an inverted-U function, as well, namely $h(x) \overset{\text{def}}{=} \phi x e^{-\kappa x}$ ($\phi, \kappa > 0$).

Because the kinetics of the nAChRs are faster than those of the neurons, we will assume Eq. 3 is at steady state. Similarly, we will assume the GINs' response to activation of nAChRs is faster than the CINs response to activation by GINs, or in other words that $\tau_G \ll 1$. Therefore, we will use the steady state solution of Eq. 2, as well. Thus, taken together we can reduce the dynamics of the CINs to

$$\frac{dC}{dt} = -C + \left[ I_c - \beta I_G - \beta h(C) - D \right]_+ \tag{4}$$

We will be interested in the parameter regime where the argument of the threshold-linear function is positive, hence Eq. 4 can be simplified to

$$\frac{dC}{dt} = A - C - \beta h(C) - D \tag{5a}$$

where $A = I_C - \beta I_G$ (Fig. 4b, light blue background). The equation for the $D$ is

$$\frac{dD}{dt} = -D + \gamma h(\sigma C) \tag{5b}$$

because DA is activated by the nAChRs which are activated by the CINs (Fig. 4c). $\gamma > 0$ represents the gain of the nAChR activation of the DA fibers, and $\sigma > 0$ is related to the "affinity" of the nAChRs on DA fibers to the activation of CIN (i.e., $\sigma = 1$ means that the nAChRs on GINs and on DA respond with identical sensitivity to CIN activity).

As explained above, we assume that the CINs can be represented by a spatially extended variable, $u(x,t)$ (for simplicity we conduct the analysis in one dimension). DA is also represented as a spatially extended variable, $v(x,t)$ (Fig. 4c). When diffusion is added, Eqs. 5a and b become the following coupled partial differential equations (PDEs) that describe the local coupling of CINs and DA fibers in the striatum

$$\frac{\partial u}{\partial t} = D_u \frac{\partial^2 u}{\partial x^2} + f_i(u) - v \tag{6a}$$

$$\frac{\partial v}{\partial t} = D_v \frac{\partial^2 v}{\partial x^2} + g_i(u) - v \tag{6b}$$

where $f_1(u) = A - u - \beta h(u)$ and $g_1(u) = \gamma h(\sigma u)$. We will refer to this model (for which the subscript $i = 1$) as the "full model". $D_u$ and $D_v$ are the effective diffusion coefficient of $u$ and $v$, respectively. In this model, we are not considering physical diffusion of ACh and DA, but rather the "diffusion" of the activity. We will consider two regimes in our analysis. In the first we will assume that $D_v = 0$, which corresponds to a regime where activation of nAChRs on DA fibers, can induce local activation of these fibers (and presumably release of DA), but does not cause (electrical) activity to propagate throughout the DA axonal arbor. This regime is amenable to analysis. In the second regime $D_v$ is allowed to be non-zero, which corresponds to electrical activity propagating throughout DA axonal arbor, as was recently demonstrated[21,25]. We shall see that $D_v > 1$ does not qualitatively alter the traveling wave solutions. However, it can give rise under certain conditions to the appearance of Turing patterns[33] of isolated hills of activity.

In order to facilitate the analysis of Eqs. 6a and b, we will also consider polynomial functions in place of $f_1(u)$ and $g_1(u)$ that will preserve the local geometry of the nullclines in the $(u,v)$ phase plane. All simulations were run on XPPAUT[67] (See Supplementary Code).

## Simultaneous advancing and receding traveling waves of CIN and DA activity

Traveling wave solutions in an extended medium arise when the dynamics of the medium are bistable such that one region of the medium is at one stable fixed-point solution of the diffusion-less system (i.e., $D_u = D_v = 0$ in Eqs. 6a and b), while the other is at the other stable fixed-point. Then, when diffusion is re-instated, a traveling wave can form as a (moving) boundary between these two solutions. The velocity (direction and speed) of the wave is determined by the parameters of the equations that will determine which fixed point will eventually win-over the media (the wave solution and its direction will depend on initial conditions, as well). Bi-stability arises in Eqs. 6a and b due to the inverted-N shaped of $f_1(u) = A - u - \beta h(u)$ which is inherited from the inverted-U shape of $h(u)$ (Fig. 5). The two stable fixed points can be identified by the flow field around them. We will consider a regime where the nAChRs on DA fibers and GINs exhibit a similar sensitivity to ACh concentration (i.e., $\sigma$ is close to unity). Note that in this region the two-stable fixed-points of the dynamics—in the absence of diffusion—are arranged geometrically such that the left stable fixed-point, $(u_1, v_1)$, is a state of high DA and low CIN, whereas the right stable fixed-point, $(u_3, v_3)$, is that of low DA and high CIN (Fig. 5a). In this case, when diffusion is introduced, the traveling wavefront of one variable advances while the other one recedes. The velocities of both wavefronts are equal, and depend on the parameters of $f_1(u)$ and $g_1(u)$ and on the diffusion coefficients. For example, increasing $\beta$, which represents increasing the gain of the nAChRs on the GINs, causes the CIN profile to transition from expanding to receding (Fig. 5b and Supplementary Fig. 3a). The parameter $A = I_C - \beta I_G$ (Eqs. 4 and 5a) can also control the direction of the wave propagation (Supplementary Fig. 3b) or terminate them, indicating that global changes in activity levels of (or common inputs to) the interneurons (see Eqs. 1 and 2) can affect the reciprocal dynamics between CINs and DA axons.

To gain a fuller understanding of the behavior of the system, we will consider an analytically tractable version of Eqs. 6a and b. In this case, $f_2(u) = u(1-u)(u-s) + a^2(1-u)$ is an inverted-N shaped third-degree polynomial, and $g_2(u) = bu(1-u)$ is an inverted-U shaped second-degree polynomial (Fig. 6a). In this model, we define $a_m = (s+b)/2$ and require that $0 < a < a_m < 0.5$, which guarantees that, in the absence of diffusion, the system's two stable fixed-points are

$$\text{Low CIN, high DA}: u_1 = a_m - \sqrt{a_m^2 - a^2}; \ v_1 = bu_1(1 - u_1) \tag{7a}$$

and

$$\text{High CIN, low DA}: u_3 = 1; \ v_3 = 0 \tag{7b}$$

Thus, the purpose of the parameter $a$ in $f_2(u)$ is to create the high DA solution (because $v_1 > v_3$ if and only if $a > 0$). In this case, we search for a traveling wave solution for Eqs. 6a and b, with velocity $c$, of the form

$$u(x,t) = U(z) = U(x - ct) \tag{8a}$$

$$v(x,t) = V(z) = V(x - ct) \tag{8b}$$

In this case, the PDE system (Eqs. 6a and b) transforms into a pair of ordinary differential equations

$$-cU' + cV' = D_u U'' + (1 - U)(U - u_1)(U - u_2) \tag{9a}$$

$$-cV' = D_v V'' + bU(1 - U) - V \tag{9b}$$

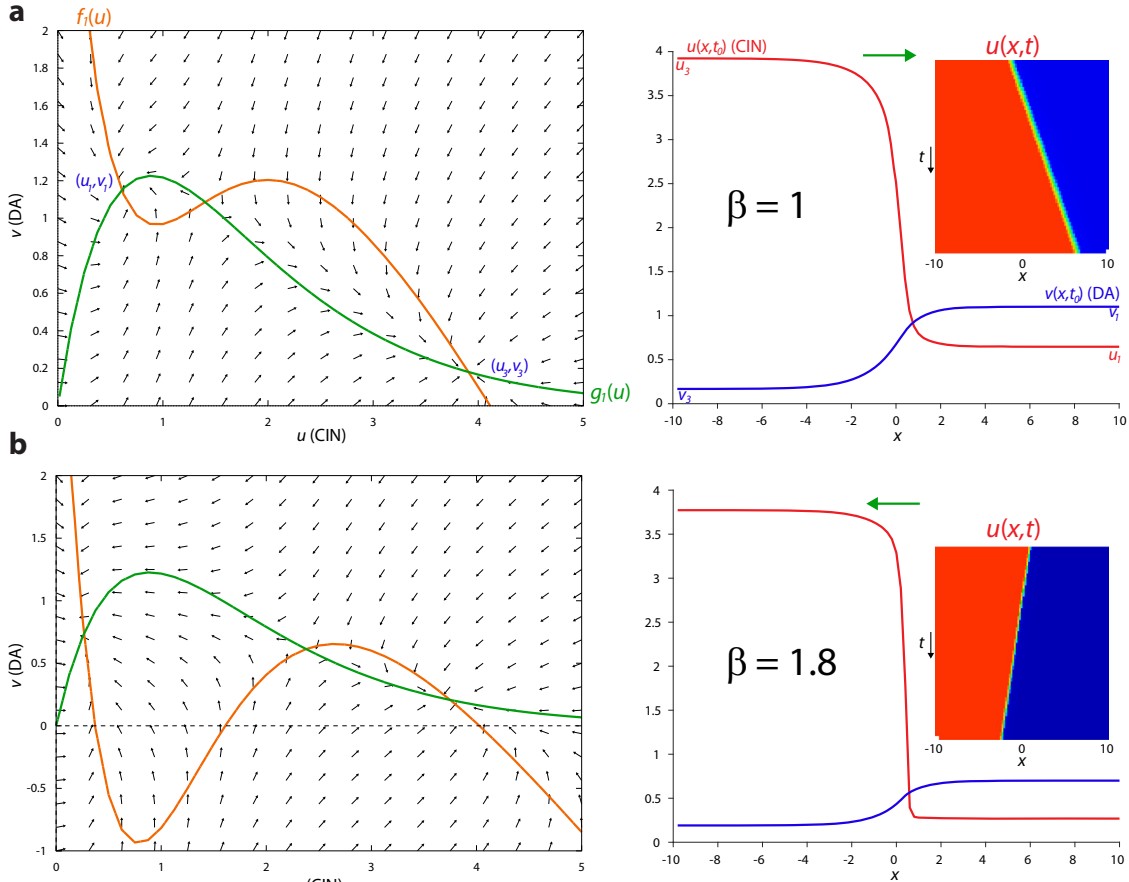

**Fig. 5 | Simultaneous advancing and receding waves of DA and CIN activity in AIRD model. a** Left: Phase plane. The diffusion-less system's nullclines intersect at 3 fixed-points, such that $(u_1,v_1)$ and $(u_3,v_3)$ (where $u_3 > u_1$ and $v_3 < v_1$) are stable, as is evident from the flow fields. When $\beta = 1$ the area between the nullclines that is adjacent to $(u_1,v_1)$ is smaller than the area that is adjacent to $(u_3,v_3)$. Right: As a result the traveling waves of $u(x,t)$ and $v(x,t)$ move such that $(u_3,v_3)$ wins over, meaning that the $u$ (CIN) wave advances and the $v$ (DA) wave recedes. Inset: space-time plot of $u(x,t)$. **b** Same as **a**, except that when $\beta = 1.8$, the area adjacent to $(u_1,v_1)$ is larger and the traveling waves are such that the $u$ wave recedes and the $v$ wave advances. Other parameters: $A = 4.2$, $\sigma = 0.75$, $\kappa = 1.5$, $\gamma = 0.47$, $\phi = 10$, $D_u = 0.02$, $D_v = 1$.

where $u_2 = a_m + \sqrt{a_m^2 - a^2}$, and the primes denote differentiation with respect to $z$.

We can gain insight into the behavior of this system by considering the case of $D_v = 0$ (which corresponds to conditions where DA fibers can be locally activated by ACh, but this activation cannot propagate throughout the DA fibers). In this case, we can analytically solve the spatial profile of a standing wave (i.e., a traveling wave with velocity $c = 0$), which is given by (Fig. 6b)

$$U(z) = u_1 + \frac{(1-u_1)}{2}\left[1 \mp \tanh\left(\frac{[1-u_1]z}{2\sqrt{2D_u}}\right)\right] \quad (10a)$$

and

$$V(z) = b\left\{\frac{u_1(1-u_1)}{2}\left[1 \pm \tanh\left(\frac{[1-u_1]z}{2\sqrt{2D_u}}\right)\right] + \frac{(1-u_1)^2}{4}\left[1 - \tanh^2\left(\frac{[1-u_1]z}{2\sqrt{2D_u}}\right)\right]\right\} \quad (10b)$$

with the condition that $c = 0$, where

$$c = \sqrt{\frac{D_u}{2}}\left(3\sqrt{a_m^2 - a^2} + a_m - 1\right) \quad (11)$$

The overshoot in the profile of $V(z)$, results from the fact that when $c = 0$, Eq. 9b can be rewritten simply as $V = bU(1-U)$, so that the

parametric trajectory $(U(z), V(z))$ runs along the green curve in Fig. 6a. Following the curve from $(u_3, v_3)$ to $(u_1, v_1)$, demonstrates that it overshoots the value of $v_1$ before approaching $(u_1, v_1)$. Similar waveforms and traveling wave behavior can occur for $u(x,t)$ and $v(x,t)$ in the full model with $f_1(u)$ and $g_1(u)$ when $D_v = 0$ (Supplementary Fig. 3c).

Note that the condition $3\sqrt{a_m^2 - a^2} + a_m - 1 = 0$ can also be derived by requiring that the integral $\int_{u_1}^{u_3}[f_2(u) - g_2(u)]du = 0$, which is a general result for standing waves in bistable systems[68–70]. Geometrically this means that the areas confined by the two nullclines between each of the stable fixed points and the unstable fixed-point $(u_2)$ need to be equal to each other (purple and green areas in Fig. 6c). The condition $c = 0$ can be rewritten as the curve (Fig. 6d, dashed line)

$$b = \frac{3}{4}\sqrt{1 + 8a^2} - \frac{1}{4} - s \quad (12)$$

which splits the system's phase diagram (Fig. 6d) into two regions: one (bottom right in Fig. 6d) where the $u$ wave advances and $v$ wave recedes (i.e. the uniform solution Eq. 7b, wins out); and the other (top left in Fig. 6d) where $u$ recedes and $v$ advances (i.e. the uniform solution Eq. 7a, wins out) (Supplementary Fig. 4). The values of the function $c(a,b;s,D_u)$ (Eq. 11) are color coded on the $b$-$a$ phase diagram, even though they do not, in general, dictate the correct velocity for each point in the phase diagram. However, for small absolute values of $c$ (and particularly if $b \ll 1$) the term $cV'$ in Eqs. 9 will be negligible, so that

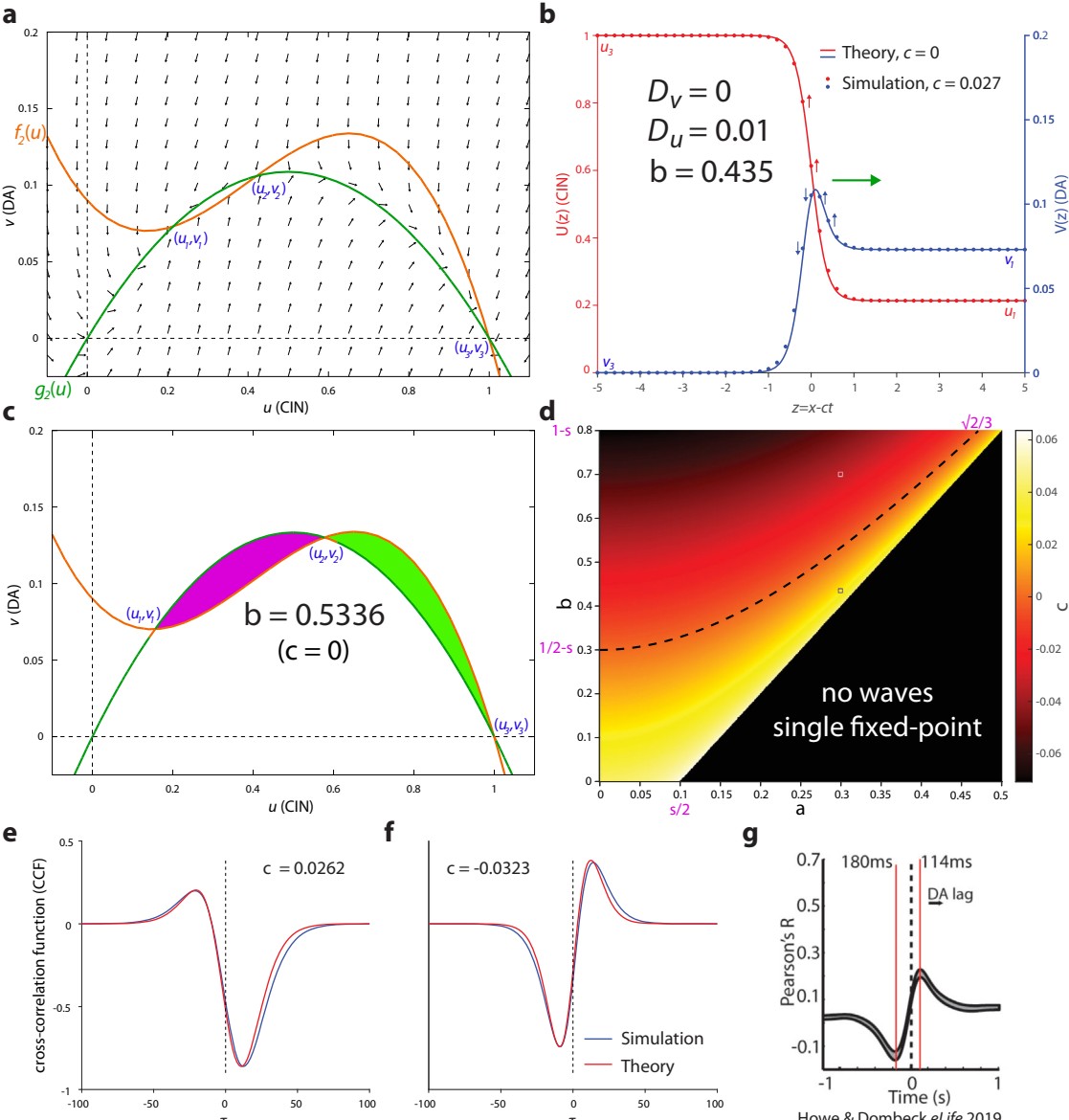

**Fig. 6 | Tractable AIRD model produces various temporal correlation structures between DA and CIN depending on the structure and direction of the traveling waves. a** Left: Phase plane. The diffusion-less system's nullclines intersect at 3 fixed-points, such that $(u_1,v_1)$ and $(u_3,v_3)$ (where $u_3 > u_1$ and $v_3 < v_1$) are stable, as is evident from the flow fields. **b** When diffusion of $u$ is introduced (using the same parameters as in (**a**)), traveling waves form such that $u$ advances and $v$ recedes, with velocity $c$ that depends on these parameters according to Eq. 11. Analytical solution of waveform for $c = 0$ matches the simulated shape for small magnitudes of $c$. **c** A standing wave of $c = 0$ forms for parameters where the (purple and green) areas adjacent to the two stable fixed-points are equal. **d** Phase diagram. Dotted line indicates $c = 0$. Velocity according to Eq. 11 is color coded. **e** Cross-correlation Function (CCF) between $u(x_0,t)$ and $v(x_0,t)$ as the traveling waves traverse $x_0$ for a positive value of $c$ [blue−simulation; red−theory (Eq. 14)] for parameters corresponding to the white circle in (**d**). **f** Same as (**e**), except that the value of $c$ is negative and corresponds to the black circle in (**d**). **g** Reproduction of a recent empirical CCF−between CGaMP6 signals recorded with fiber photometry from DA and CIN striatal neuropil[20]−that resembles the shape of the CCF in (**f**). Other parameters: $a = 0.3$, $s = 0.2$ (unless stated otherwise).

the solution described in Eqs. 10 will be still approximately valid (Fig. 6b, "Theory" vs. "Simulation").

## Temporal relationship between DA and CIN activity

A standard experimental method to characterize the temporal relationship between two signals is to calculate the temporal cross-correlation function (CCF) between them. If we consider any point $x_0$, we can intuit how the two signals will change there in time. As the waveforms move to the right, i.e. $c > 0$ (Fig. 6b), the points along the wavefronts will change as indicated in the arrows (and this can be viewed directly in Supplementary Fig. 4). Formally, this means that that the change in activity at that point is given by the temporal derivative of the signals, i.e., $-cU'(x_0-ct)$ and $-cV'(x_0-ct)$. Thus, the CCF can be

calculated as

$$C(\tau) = \int_{-\infty}^{\infty} U'(x_0 - ct)V'(x_0 - c[t+\tau])dt \tag{13}$$

which, from translational symmetry of the traveling wave, is independent of $x_0$.

If we consider $x_0 = 0$ in Fig. 6b, as the CIN signal there increases (upward red arrows), it will be positively correlated with the DA signals that are to its right (upward blue arrows). This means that the DA signal leads the CIN signal and the maximal correlation will be attained at a negative time delay, where DA precedes CINs (e.g., Fig. 6e, blue curve). Note that in this case, there is a region at the leading edge of wavefront

where the CIN and DA activities are elevated together. This region represents the ability of CINs to drive DA release by activating nAChRs on DA axons (Fig. 2)[21–23,36]. Conversely, if the wavefronts move to the left, i.e., $c < 0$, all the arrows will point in the opposite direction. In this case, the downward change in the CIN signal will be positively correlated with the lagging DA signals, which means that the maximal correlation will be attained at a positive delay, where DA lags behind CIN (Fig. 6f, blue curve).

The CCF can be calculated analytically for the wave solutions in Eq. 10. Because $V(z)$ (Eq. 10b) is composed of two terms, the CCF is also composed of two terms, one symmetric, $C_S(\tau)$, and another anti-symmetric, $C_A(\tau)$:

$$C(\tau) = -u_1 C_S(\mu\tau) - \frac{1-u_1}{2} C_A(\mu\tau) \qquad (14)$$

where

$$C_S(\tau) = \frac{\tau \coth(\tau) - 1}{2\sinh^2(\tau)} \qquad (15a)$$

$$C_A(\tau) = \frac{d}{d\tau} C_S(\tau) \qquad (15b)$$

and

$$\mu = \frac{[1-u_1]c}{2\sqrt{2D_u}} = \left[ 3(a_m^2 - a^2) - (1 - a_m)^2 + 2(1 - a_m)\sqrt{a_m^2 - a^2} \right]/4 \qquad (16)$$

The theoretical calculation (Fig. 6e, f, red curves) closely resembles the numerical solution (blue curves). Intriguingly, the case where the CIN waveform recedes generates a CCF that resembles the empirical CCF measured recently between $Ca^{2+}$ signals from CINs and striatal DA fibers that expressed GECIs, and were recorded with fiber photometry[20] (Fig. 6g).

Note that the width of the CCF, determined by the parameter μ (Eq. 16), is strictly a function of the parameters of the diffusion-less system (i.e., a function of the parameters of the local interaction between CINs and DA axons), and are independent of the diffusion coefficient, $D_u$, that affects the speed of the wave. When $D_u$ is large, the wave is both faster and has a spatially broader interface. Conversely, when $D_u$ is small, the wave is slower and has a narrower interface. Thus, in this case of a single diffusion coefficient ($D_v = 0$), these two effects (speed and width of interface) cancel out, causing the functional shape of the CCF to be the same for a given temporal delay $\tau$, independently of $D_u$.

## Turing instability can trigger traveling waves of CIN and DA activity

For $D_v > 0$, numerical simulations show that the traveling waves are still the solution to the system, but their shape is not given by Eq. 10. Moreover, Eq. 11 is no longer valid and the reversal of the velocity of the traveling wave occurs elsewhere in the phase diagram (Fig. 6c). Thus, whether $D_v$ is zero or non-zero does not qualitatively alter the traveling wave solutions, indicating that striatal waves can occur whether or not nAChRs trigger a local traveling AP in DA axons, as was recently shown[21,25]. Interestingly, only for the case of non-zero $D_v$, there exists a parameter regime where the uniform solution ($u_1$, $v_1$) (Eq. 7a) loses its stability through a Turing instability (Methods). Spatial patterns (of a particular spatial scale, determined by the parameters) form spontaneously and transiently and trigger a traveling wave of advancing $u$ and receding $v$, i.e., ($u_3$,$v_3$) (Eq. 7b) wins out (Fig. 7a, b and Supplementary Fig. 5). As explained in the Methods, the parameter regime where this is possible (provided $D_u/D_v$ is sufficiently small) is determined by the inequality $A_{11} > 0$, which is an entry in the stability

matrix of the diffusion-less system in the vicinity of one of the fixed-points (Eq. 7a), and which translates into (Fig. 7c, yellow region):

$$b < \frac{2}{3} \frac{(s + \frac{1}{4})^2 + 3(\frac{5}{16} + a^2) - (1+s)\sqrt{(s-\frac{1}{2})^2 + 3(\frac{1}{4} - a^2)}}{1 + s - \sqrt{(s-\frac{1}{2})^2 + 3(\frac{1}{4} - a^2)}} - s \qquad (17)$$

## Simultaneous advancing or receding traveling waves of CIN and DA activity

In the models we analyzed above, while the wavefronts of ACh and DA overlapped and traveled together, one receded while the other advanced. This resulted from the geometry of the intersections between the nullclines which gave rise to two stable fixed-points of high $v$ (DA) and low $u$ (ACh) or vice versa. In order for the two traveling waves to advance together or recede together the nullclines need to intersect such that they give rise to one fixed-point of low $u$ and low $v$ and another of high $u$ and high $v$. This can be attained in the full model [with $f_1(u)$ and $g_1(u)$] if $\sigma \ll 1$ (Fig. 8). Physiologically, this represents the case where the nAChRs on the GINs have a substantially higher affinity to ACh, than the nAChRs on the DA fibers. As expected, for appropriate parameters the traveling waves of DA and ACh in this parameter regime advance or recede together (Fig. 8 and Supplementary Fig. 7).

Interestingly, if we choose parameters where the stable high $u$ and high $v$ fixed-point of the diffusion-less model occurs where the slope of $f_1(u)$ is positive (Fig. 9a), the full model gives rise to receding traveling waves that leave a spatial pattern in their wake (Fig. 9b, c and Supplementary Fig. 8).

To gain a fuller, analytical understanding of this parameter regime and of the pattern formation, we once again replace $f_1(u)$ and $g_1(u)$ with tractable polynomial models: $f_3(u) = u(1-u)(u-s)$ and $g_3(u) = bu$, with $0 < s < 1$ and $b > 0$. This model (Fig. 10) is known as the Fitzhugh–Nagumo model[71].

In the absence of diffusion, this system has two stable fixed points:

$$\text{Low CIN, low DA}: u_0 = 0; \quad v_0 = 0 \qquad (18a)$$

and

$$\text{High CIN, high DA}: u_+ = \frac{1}{2}\left(1 + s + 2\sqrt{b_{max} - b}\right); v_+ = bu_+ \qquad (18b)$$

with $b_{max} \stackrel{\text{def}}{=} (1-s)^2/4$, so that the condition for the existence of the second stable fixed point is that $b \le b_{max}$.

Simulating the system with diffusion (i.e., $D_u = 0.1$ and $D_v = 1$) demonstrated that the two wavefronts advanced or receded together (Fig. 10a, b and Supplementary Fig. 9). Repeating the traveling wave analysis (Eqs. 8a and b) for this model, with $D_v = 0$, we find traveling wave solution whose velocity is given by

$$c = \frac{\sqrt{D_u}}{2\sqrt{2}}\left[6\sqrt{b_{max} - b} - (1+s)\right] \qquad (19)$$

For the case where $c = 0$, we can derive a precise waveform solution for the standing wave

$$U(z) = \frac{u_+}{2}\left[1 \pm \tanh\left(\frac{u_+ z}{2\sqrt{2D_u}}\right)\right]; \quad V(z) = bU(z) \qquad (20)$$

The condition $c = 0$, can be also derived from the requirement $\int_0^{u_+}[f_3(u) - g_3(u)]\,du = 0$ which yields

$$b = \frac{2}{9}(1+s)^2 - s \qquad (21)$$

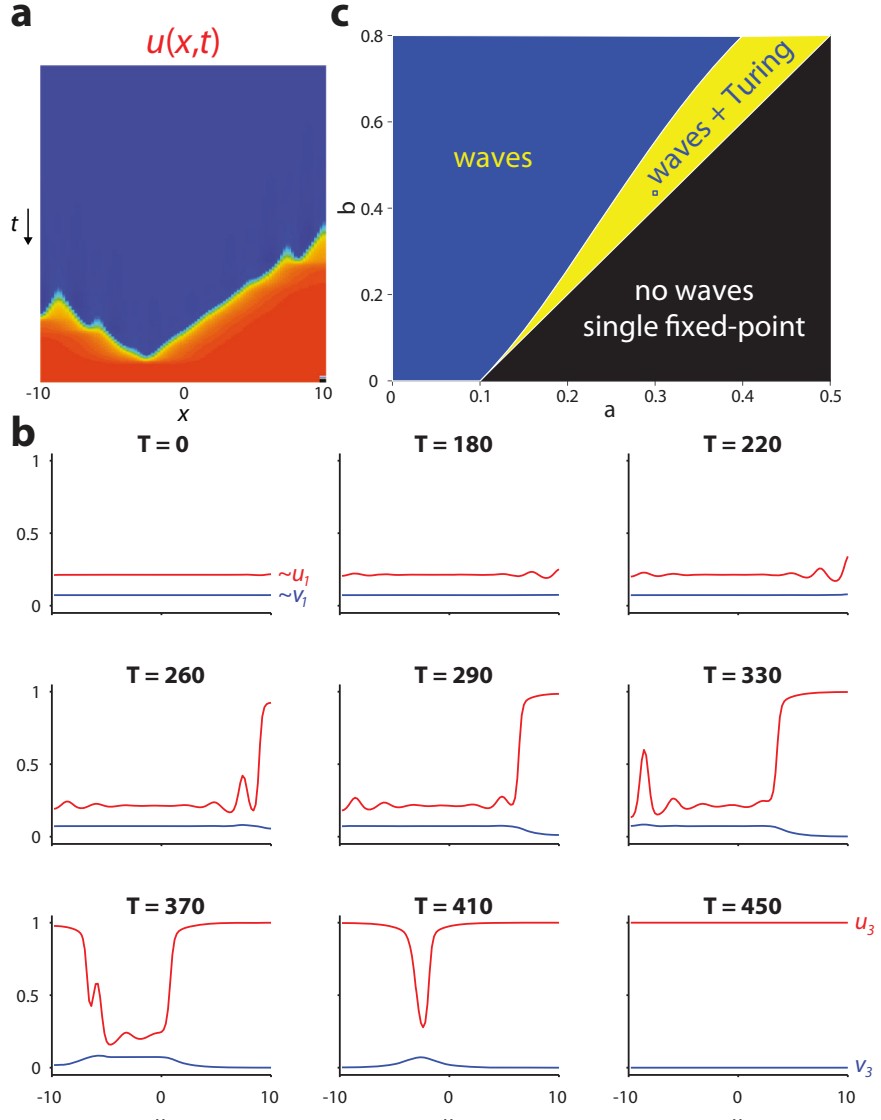

**Fig. 7 | Turing instability in the tractable AIRD model. a** Space-time plot of $u(x,t)$, and **b** series of snapshots describing the "spontaneous" destabilization by diffusion of the low $u$ (CIN)/high $v$ (DA) state, through a Turing instability. Transient localized "hills of activity" form, and they are overtaken by traveling waves that drive the system to the high $u$ (CIN)/low $v$ (DA) state. Parameters: $a = 0.3$, $b = 0.435$, $s = 0.2$, $D_u = 0.06$, $D_v = 1$. **c** Indication of the region in the phase diagram where these transient Turing patterns can form (provided $D_u/D_v$ is small enough).

This curve (Eq. 21) divides the phase diagram into two regions: one where the waveforms for $u$ and $v$ recede (Fig. 10c, red region), and one were they advance (Fig. 10c, blue region). Calculation of CCF for Eq. 20 yields a positive symmetric $C(\tau) = C_S(\lambda\tau)$ (see Eq. 15a) with

$$\lambda = \frac{u_+ c}{2\sqrt{2D_u}} = \left[ s - b + 2(b_{max} - b) + (1+s)\sqrt{b_{max} - b} \right]/4 \quad (22)$$

The symmetric CCF is maintained also when $D_v > 0$ (Fig. 10d). Thus, this model only gives rise to in-phase synchrony between CIN and DA. The width of the CCF depends on the velocity of the waves (Supplementary Fig. 9): when the wave is faster (e.g., for $s = 0.25$), the CCF is narrower (Fig. 10d).

In the models discussed in the previous sections, while the Turing instability could give rise to transient spatial patterns it was impossible for these patterns to stabilize, because the two variables opposed each other at every given point: where one tended to a low value the other tended to a high value and vice versa, so this drove them apart and dissipated the initial destabilizing transient spatial pattern. In contrast, in the current models, where the two variables tend to high and low values together, it is possible that Turing patterns will stabilize with the "long-range" inhibition by DA separating space into localized regions that are elevated by the "short-range" excitation by CINs.

Conducting the stability analysis near the activated state (Eq. 18b), the condition for a Turing instability (provided Eqs. 25a and b in the "Methods" is fulfilled) is (Fig. 10c, yellow region)

$$b > \frac{1}{9}\left[ (1+s)^2 - 6s + (1+s)\sqrt{(1+s)^2 - 3s} \right] \quad (23)$$

As expected, in this parameter regime we can observe receding waves that leave either transient or stable Turing patterns of localized activity in their wake (Fig. 10e and Supplementary Fig. 10). This analysis

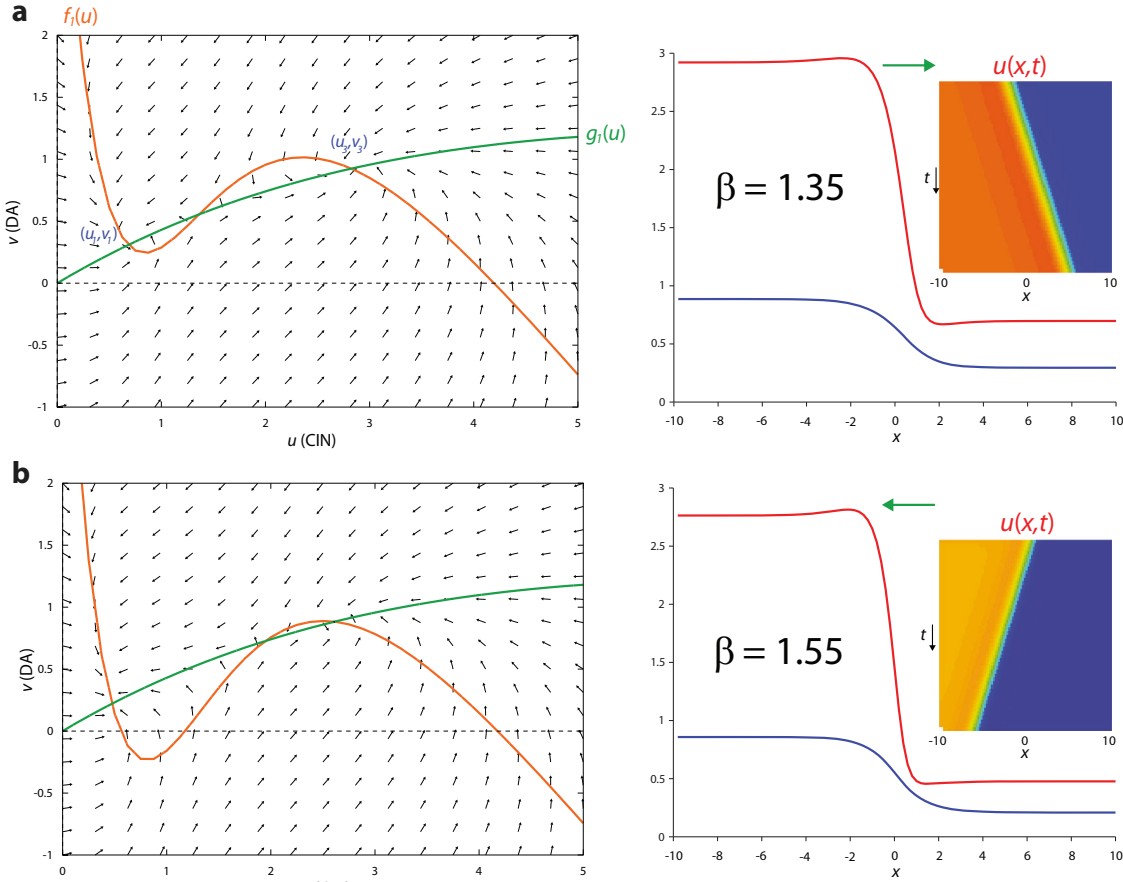

**Fig. 8 | Simultaneously advancing or receding traveling waves in full model [with $f_1(u)$ and $g_1(u)$]. a** Left: Phase plane. Setting $\sigma = 0.1$ horizontally stretches the green nullcline (which is equivalent to reducing the affinity of the nAChRs on the DA fibers to ACh) causing it to intersect the orange nullcline at three fixed-points, with the two stable fixed-points arranged such that $u_3 > u_1$ and $v_3 > v_1$. When $\beta = 1.35$, the area between the nullclines adjacent to $(u_3, v_3)$ is larger than the area adjacent to $(u_1,$ $v_1)$. Right: $u$ and $v$ form traveling waves where both fronts advance (rightwards) together. Inset: space-time plot. **b** Left: Same as panel a, except that setting $\beta = 1.55$ causes the area between the nullclines adjacent to $(u_1, v_1)$ to be larger. Right: Same as (**a**), except that now the fronts recede (leftwards) together. Other parameters: $A = 4.3$, $\kappa = 1.5$, $\gamma = 0.47$, $\phi = 10$, $D_u = 0.2$, $D_v = 1$.

also provides an explanation for the patterns observed in the full model (Fig. 9b, c and Supplementary Fig. 8). In particular, we found that the patterns only occur in the full model $[f_1(u), g_1(u)]$ when the nullclines intersect such that unstable fixed point is close to the fixed point with the high values of $u_3$ and $v_3$. This corresponds to the yellow band in Fig. 10c in the tractable model $[f_3(u), g_3(u)]$ being adjacent to the region where that two fixed points are close to merging—which is close to the region where only a uniform trivial solution exists— namely, when $b$ is close to $b_{max}$. In these narrow regions there can occur a stable fixed point $(u_+, v_+)$ where the slopes of $f_1(u)$ (Fig. 9a) or $f_3(u)$ are still positive.

## Local CIN–DA interaction does not preclude brief, localized and diffusive patterns
Whereas the focus of our analysis has been on the regime of traveling waves and their properties, there are certainly initial conditions in which waves are not triggered in our models. For example, local activation of DA fibers or CINs from a homogeneous state can either succeed or fail to trigger traveling waves (Supplementary Fig. 6). Thus, when waves fail to form (particularly when the input is weak), the system will exhibit what looks like a local diffusive response, which may be only partially affected by the nonlinearities. Similarly, when we propose that similar nonlinear mechanisms might exists in the DS, we are not arguing that every spatiotemporal pattern observed is a wave, in accordance to what we observed empirically (Fig. 1). Instead, we

argue that the appearance of waves (interspersed among localized events) per se can be indicative of a local interaction between CINs and DA axons that is capable of supporting waves and other nonlinear spatiotemporal patterns.

## Summary of model predictions
Several testable predictions can be derived from the model. In addition to predicting the existence of ACh traveling waves intertwined with DA waves, the model predicts that the local interaction between DA axons and CINs is sufficient to generate waves. External input is not necessary, suggesting that traveling waves can occur even when the animal is in a state of quiet rest, as was observed both with respect to DA[8] and ACh (Fig. 1) waves. Two other predictions are that, under conditions of no external input, DA–ACh interactions (as formulated in the model) are necessary to generate waves, and that APs in individual CINs cause DA release from axons (without requiring CIN synchrony). Conversely, blocking ACh receptors should reduce both DA and ACh waves. Similarly, enhancing or reducing DA signaling would alter DA and ACh wave-like behavior. The model also predicts that the direction of the wave motion can be reversed by a sudden increase in the feedforward activation of the CINs. Under certain conditions, isolated hills of activity can occur. Although we have validated the first few of these predictions, further experimental work would be required to verify the latter predictions of the model.

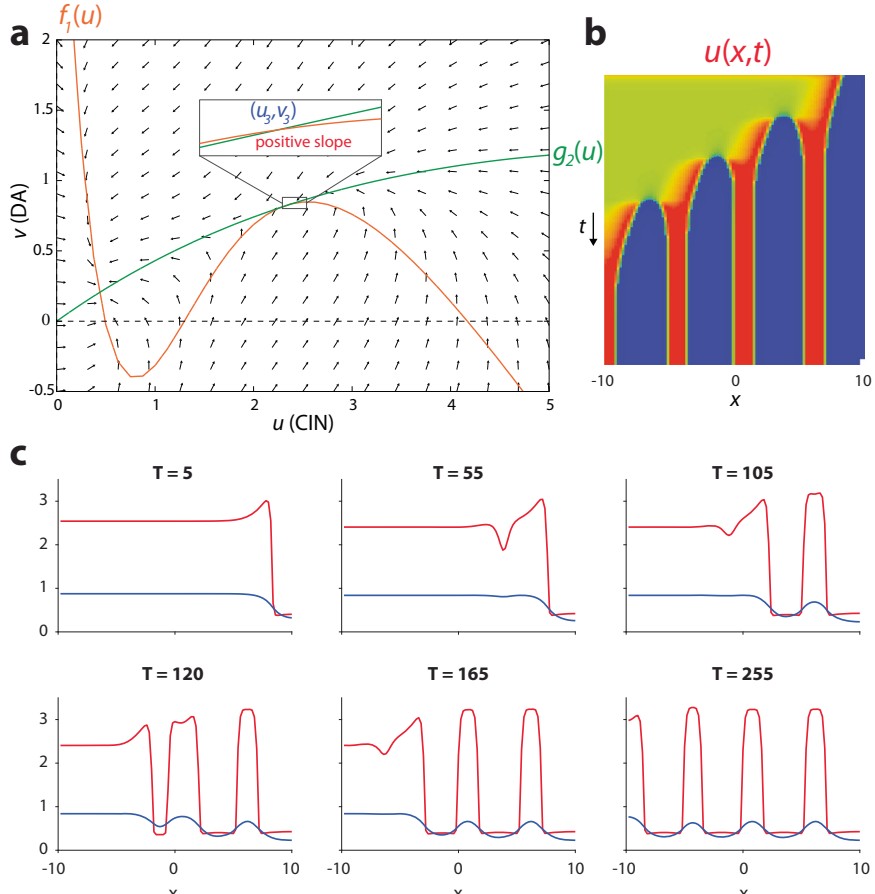

**Fig. 9 | Stable Turing patterns form in the wake of receding traveling waves in full model [with $f_1(u)$ and $g_1(u)$]. a** Phase plane. When $\beta = 1.62$ the area between the nullclines adjacent to $(u_3, v_3)$ is very small, the two nullclines cross with positive slopes (inset) that fulfill the condition for a stable Turing bifurcation (see "Methods"). **b** In this parameter regime, the space-time plots reveal a receding traveling wave that leaves stationary and localized peaks of $u$ standing **c** Snapshots of the receding wavefronts reveal how they leave localized hills of activity in its wake, with the spatial spread of the $v$ (DA) being wider than that of $u$ (CIN). Other parameters: $A = 4.3$, $\sigma = 0.1$, $\kappa = 1.5$, $\gamma = 0.5$, $\phi = 10$, $D_u = 0.01$, $D_v = 1$.

## Discussion

### Empirical evidence for wave activity in striatal ACh release and for the control of the spatiotemporal pattern of DA release by CINs

In this study, we have reported three experimental findings. First, we demonstrated waves in striatal ACh release imaged in head-fixed mice. Second, we demonstrated that the spatial extent of striatal DA release triggered with bipolar electrode stimulation depends on nAChRs that extend the range over which DA release occurs by several hundred micrometers, in agreement with a very recent study[21]. Finally, we demonstrated in acute striatal slices that APs elicited in individual CINs can produce local DA release, albeit infrequently. The localized nature of the release—in some cases we could only detect DA in close proximity to a single stretch of axon but not a few 10 s of microns away (Fig. 3)—and the low yield of this experiment make clear why it has not previously been detected using fast-scan cyclic voltammetry[22,23] or other methods[21,25].

Collectively, these findings formed the basis for the formulation of the theoretical modeling part of this study. The fact that both DA and CIN release form spatiotemporal wave-like activity, in conjunction with the fact the CINs can extend the spatial scale of DA release through nAChR activation, prompted us to hypothesize that it is the CINs that drive simultaneous waves of DA and CIN activity. We, therefore, investigated a common mechanism that could explain the existence of both spatial waves, based on the idea that the driver is the CINs. Moreover, we hypothesized that the dynamic organization of

these waves occurs locally in the striatum by direct interaction between CINs and DA fibers. A natural mechanism that fits these constraints is an AIRD system, in which the CINs are the activators and DA fibers are the inhibitors. Thus, we developed a dynamical model of coupling between CINs and DA fibers that relies on a physiologically-inspired model of their local interaction. We then supplemented this local interaction with a model of diffusion within a syncytia of cholinergic and dopaminergic fibers. We demonstrated that this system can give rise to traveling wave solutions, and analyzed the parameters space of tractable systems that are essentially equivalent to the physiologically-inspired model, because they maintain the local geometry of nullclines and fixed-points.

### Nicotinic gain and the degree of interneuronal activation determine the direction of the traveling waves

The direction and velocity of the traveling wave solutions are determined by the parameters of the system. In our models, because of the bi-stability in the CIN dynamics, which gives rise to their putative self-excitation (see below), it is the CINs that drive the traveling waves, whereas the DA is the follower. Therefore, the strength of the CIN coupling to the DA fiber in the models is a critical parameter. In the tractable models, this is represented by the parameter $b$ (which is roughly equivalent to the product of $\gamma$ and $\sigma$, in the full physiologically inspired model). When considering the phase diagrams of both models (Figs. 6d and 10c) it is evident that increasing $b$ causes the direction of the traveling CIN wave to switch from advancing to receding. In the

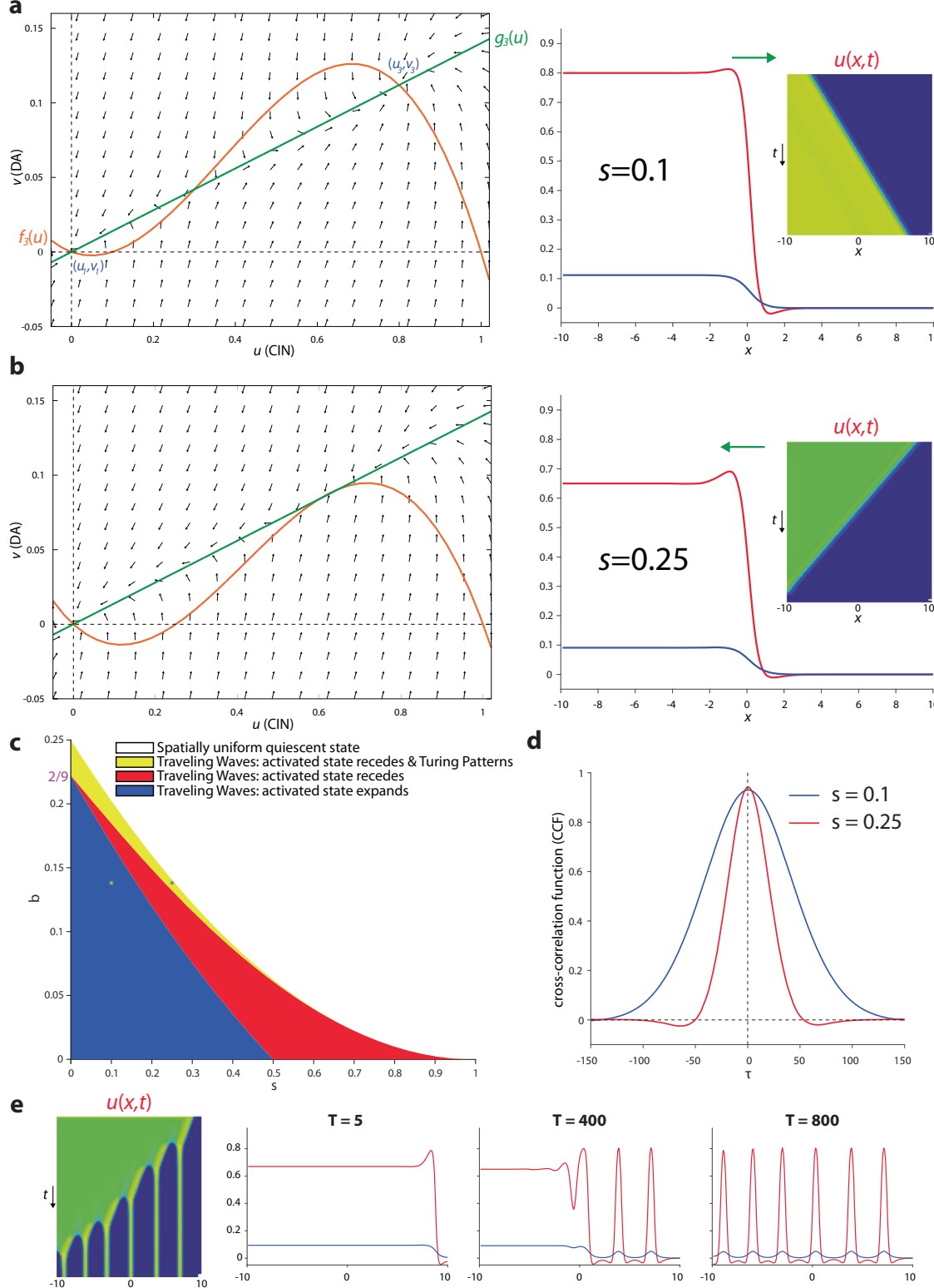

**Fig. 10 | Fitzhugh–Nagumo (FN) model for simultaneous advancing and receding traveling waves and Turing patterns [with $f_3(u)$ and $g_3(u)$].** **a** Phase plane. Arrangement (Left) of fixed-points (e.g., $u_+ > u_0$ and $v_+ > v_0$) and nullclines [e.g., area between nullclines adjacent to $(u_+,v_+)$ is larger] that give rise to simultaneously expanding "activated" states (Right). **b** Same as panel a, except that area between nullclines adjacent to $(u_0,v_0)$ is larger, giving rise to simultaneously receding activated states Insets: space-time plots. **c** Phase diagram of FN model,

demonstrates that it supports both advancing and receding waves, and in a subset of the receding wave regime it is possible to observe Turing patterns (provided the $D_u/D_v$ is sufficiently small). **d** Cross-correlation function [of $u(x_0,t)$ and $v(x_0,t)$] during a traveling wave is always symmetrical. Parameters correspond to black and yellow asterisks in (**c**). **e** Space-time plot and snapshots of a receding wave leaving Turing patterns in its wake. Parameters correspond to black asterisk in (**c**).

models where the expansion of CIN activity is coupled to the recession of DA activity (and vice versa), this makes sense, because increasing $b-$ which signifies an increase in the gain of nAChRs on DA fibers −"strengthens the DA activation at the expense of the activation of CINs". Geometrically, increasing $b$ (in both of the tractable models) increases the area between the nullclines $f_i(u)$ and $g_i(u)$ from $u_1$ to $u_2$, so that it eventually becomes larger than the area between these curves from $u_2$ to $u_3$ (Figs. 6a, c and 10a, b). This in turn, causes the wave to switch direction to where $(u_1,v_1)$ wins out. This geometrical interpretations also helps to see why increasing $\beta$ in the full model has the same effect, even though it represents the gain of the nAChRs on the GINs (Fig. 5a, b). As $\beta$ is increased, the area between the nullclines from $u_1$ to $u_2$ increases in the same fashion, pulling the traveling wave toward $(u_1, v_1)$.

Changes in gain are essentially changes in synaptic strengths. Therefore, they may be less likely to occur on the short timescales during which the empirically observed waves change their direction (Fig. 1 and Supplementary Movies 1 and 2). Interestingly, the full model includes another parameter, $A$, (Eq. 6a) that can alter the direction of the traveling waves or terminate them. A reduction in $A$ shifts $f_1(u)$, the u-nullcline (e.g., Fig. 4a), downwards, which alters the areas between the nullclines that determine the direction of the wave. $A = I_C - \beta\ I_G$ (Eqs. 4 and 5a) represents a linear combination of the autonomous (and/or synaptic) drive to CINs (Eq. 1) and GINs (Eq. 2). Thus, if $I_G$ is increased, $A$ is reduced, which can, in turn, change the direction of the wave (Supplementary Fig. 3b). For other values of $A$ the bi-stability of the system is disrupted and a single fixed-point is established, thereby preventing and terminating wave propagation. Thus, our model does enable changes in the drive to the various striatal interneurons to alter the direction of the wave propagation or terminate it on a moment-by-moment basis. Interestingly, a recent study has shown that LTSIs can attenuate DA release in the striatum[72]. Because their action is inhibitory LTSI are not likely to drive DA waves, but they may be able to influence them. Indeed, in our model, it is precisely GINs, for whom $I_G > 0$, that can control the occurrence and direction of wave propagation. Because $I_G > 0$ represents autonomously active GINs, such as LTSIs[61-63], our model actually predicts that the activity of LTSIs should be able to determine whether ACh as well as DA waves occur in the striatum and determine their direction.

## Model assumptions: autocatalytic CINs and diffusion

The formation of wavefronts requires that the equation $f_i(u)$ (Eq. 6a) of the activator $u(x,t)$ have an inverted-N shape, because the equation for the inhibitor $g_i(u)$ (Eq. 6b) can then intersect $f_i(u)$ at 3 points, thereby creating two stable fixed points $(u_1,v_1)$ and $(u_3,v_3)$. We posited that the CINs' nullcline $f_i(u)$ inherits this shape from the known inverted-U shape of the dependence of nAChRs on ACh concentrations (which will be positively correlated with CIN activity). Because these nAChRs activate GINs[61,73,74] which are inhibitory, the "negative sign" of inhibition contributes an upright-U shaped kink into the nullcline $f_i(u)$, creating its inverted-N shape (Figs. 5a, b; 6a, c; and 10a, b). Physiologically, this means that for strong enough activation of CINs, the increase in ACh concentration (>1 mM, Fig. 4b) will lead to a deactivation of the nAChRs on the GINs, which would translate into a recurrent self-disinhibition of CINs by their own activation. This, in essence, is how we realized the self-excitatory (autocatalytic) nature of CINs in this model. If our proposed mechanism is to be taken seriously, one would need to demonstrate that α4β2 nAChRs on GINs indeed exhibit an inverted-U shape at physiological levels of striatal ACh, as was observed in *Xenopus* oocytes[64]. Similarly, in the models in which there is a high ACh/low DA fixed-point $(u_3, v_3)$, the slope of $f_i(u)$ at that fixed-point is negative. This means that the α4β2 nAChRs on the striatal DA fibers are also operating in the regime where high ACh deactivates them, which would also need to be demonstrated empirically. In the full model, this required the value of $\sigma$ to be close to

1, so that the inverted-U shape of $g_1(u)$ was visible in the same range of $u$ values (i.e., CIN activation) where the inverted-N shape of $f_1(u)$ was visible (Fig. 5). Physiologically, this means that the affinity of the α4β2 nAChRs to ACh (which we are assuming is positively correlated with the CIN activity) is similar for both the DA fibers and the GINs. In contrast, when $\sigma \ll 1$, which represents a case where the affinity of the nAChRs on the GINs is much higher than the affinity of the nAChRs on DA fibers, the kink in $f_1(u)$ occurs only in the rising part of $g_1(u)$, so that in this case CINs can only activate (but not deactivate) DA fibers.

The assumption that CINs can self-excite lies at the heart of the AIRD mechanism we propose for the generation of traveling waves. In the current state of experimental knowledge this assumption is quite tenuous. In acute striatal slices, CINs couple to one another through di-synaptic inhibition[56,57]. However, it is possible that alternative mechanisms of self-excitation exist in vivo that could support our model and give rise to traveling waves. For example, DA fibers can co-release glutamate[54,75-78]. Activation of nAChRs on DA might lead to glutamate release, which in turn would both excite CINs on a fast time scale through ionotropic glutamate receptors that would effectively provide self-excitation, and inhibit them more slowly by activation of metabotropic D2Rs[54]. Alternatively, activation of presynaptic α7 nAChRs on cortical terminals might produce self-excitation of CINs[73]. In such a scenario, activation of cortical input could introduce the bi-stability: when α7 nAChRs were activated, CINs would become autocatalytic because in this scenario their release of ACh could cause the cortical input to have a stronger excitatory influence over themselves causing self-excitation. This alternative scenario has the corollary that in the absence of cortical drive CINs cannot be autocatalytic. While all of these proposed mechanisms are speculative, they suggest predictions that can be tested experimentally, and the existence of DA and CIN waves warrants a search for some mechanistic explanation for their generation.

The diffusion term introduced into the model functions to enable the spread of activation of DA and CIN activity. We based the diffusion component on the density and intermingling of ACh and DA terminals, the affinity and proximity of DA terminal nicotinic receptors to ACh release sites, and the affinity and concentration of acetylcholinesterase (AChE), as detailed below. These considerations suggest that within a volume of a few μm³ there are sufficient numbers of ACh and DA terminals in near enough proximity to ensure interactions between them, and such contiguous volumes ensure ACh−DA interactions can occur continuously throughout the striatum. The key facts are as follows: The density of choline acetyltransferase (ChAT)-positive varicosities containing ACh vesicles is 0.20 μm⁻³ of which 10% display synaptic membrane specializations[79], giving an estimate of the density of ACh release sites of $\rho_{ACh} = 0.020$ μm⁻³. This is consistent with estimates[55] based on the quantitative electron microscopy measures of density of symmetric synapses[80] and the fraction of symmetric synapses that are cholinergic[81]. If randomly arranged this density of release sites predicts an average nearest neighbor distance of 2.04 μm using the method of Clark and Evans[82,83]. The corresponding density of DA varicosities is 0.10 μm⁻³ [6,84,85]. Assuming 17% of varicosities are active release sites[45] the density of DA release sites is about $\rho_{DA} = 0.017$ μm⁻³ and if randomly arranged the average nearest neighbor distance between DA release sites is 2.10 μm, which is consistent with the spacing of DA synapses measured in serial sections by electron microscopy[86]. Combining both ACh and DA release site densities we might expect a mean distance from ACh release sites to the nearest DA release site to be on the order of 1.6 μm. This is consistent with recent direct, super-resolution measurements indicating the distance of ACh terminals from their nearest DA axon is less than 0.5 μm in half of the cases and less than 2.0 μm in almost all cases[21]. Thus, there is a high probability of a DA release site within a sphere of 2.0 μm radius around each ACh terminal.

The spread and time course of ACh depends on diffusion from release sites and on the activity and concentration of AChE. Reported

values of $K_m$ for AChE range from $0.5 \times 10^{-4}$ to $1.4 \times 10^{-3}$ M[87,88], which are much higher than the concentration of ACh and affinity of ACh receptors. Although AChE is one of the fastest enzymes known[89,90] the effective turnover number is low at low levels of ACh. Quite high concentrations of AChE (micromolar range) would be required to hydrolyze ACh in a millisecond timeframe[91]. While such high concentrations of AChE exist in the neuromuscular junction, in the brain major AChE form in the CNS is an amphiphilic globular tetramer ($G_4$ AChE)[32,92,93] that is present at much lower concentrations on the order of 300 nM[94]. Calculations of the spatial spread and time course of ACh[55] indicate that striatal AChE would have small effects on immediate time course (ms range) of ACh concentration in the near distance (μm range) after a release event and its main effect would be on the ambient levels of ACh at longer delays (seconds) and distances (tens of μm). Moreover the majority (85%) of DA boutons contain the nicotinic ACh receptor β2 subunit[95]. The α4β2 nicotinic receptors found in vivo exhibit high agonist sensitivity ($EC_{50}$ for ACh of 0.97 μM)[96]. Thus, α4β2 nicotinic ACh receptors would be functionally effective even if located several micrometers from release sites. Consistent with this, Kramer et al.[25] found that blocking AChE did not potentiate the peak amplitude of stimulated or spontaneous ACh mediated EPSPs in DA axons but caused a later reduction in the peak amplitudes, likely due to receptor desensitization from the increased ambient concentration. They concluded that their results "suggest an arrangement whereby a fraction of CIN terminals exist in close proximity to nAChRs on dopaminergic axons."

Thus, in the model both DA and CIN axons can be conceptualized as syncytia that interact everywhere, and that once an area is activated there is potential for that activation to spread. Physical diffusion of DA and ACh from release sites is too short-ranged to produce the spread of activation observed in vivo and predicted by the model. On the other hand, the time scales of electrotonic spread are on the order of milliseconds to 10 s of milliseconds, which is presumably faster than the spreading of wavelike activity observed by ourselves and others[8]. The slower dynamics observed in vivo may reflect the rise and decay time constants of the fluorescent indicators, and thus do not rule out electrotonic spread. With all these caveats in mind, we considered two cases: one in which $D_v = 0$, which we interpret to represent a case where all the DA released is caused by CIN activation. The only activity that actually spreads is that of the CINs, and when a nAChRs is activated on a DA fiber it cannot trigger the spread of activity in the DA fibers but only trigger local release of DA. The second case we considered is that of $D_v > 0$ (we used, without loss of generality, $D_v = 1$). Here the assumption is that activity can also spread in the DA fiber syncytium[25], and that generally the diffusion of this activity is more long-ranged (as discussed below).

## Wave propagation and temporal correlations between DA and CIN activity

In the absence of direct visualization of spatiotemporal neural activity patterns, calculating the CCF between two signals is a widely used method to characterize the temporal correlation between them. Using the tractable models ($i = 2, 3$ in Eqs. 6a and b), we derived the structure of the expected CCFs between the DA and CIN signals at a given point, $x_0$ [i.e. $v(x_0, t)$ and $u(x_0, t)$, respectively]. In the case when there is a single diffusion coefficient $D_u$ in the problem (i.e., $D_v = 0$), the width of the CCF is independent of $D_u$ (Eqs. 16 and 22), meaning that even if there is a mismatch between the true underlying timescale of diffusion and the experimentally observed one, this mismatch should not strongly impact the measured temporal correlations. The tacit assumption in our calculation of the CCF is that our measurement is insensitive to the baseline value at a given region but only to the changes from that value (a reasonable assumption for many of the existing experimental techniques such as fiber photometry). Hence, the calculation of the CCF using the temporal derivatives of $U$ and $V$

(Eqs. 10 and 20). For the case where the two waves advanced or receded together, we found that the correlation is symmetric and in phase. Symmetric, in-phase CCFs have typically not been found between DA and CIN signals in the basal ganglia[20,30–32,97], arguing against coupled DA and ACh waves that advance or recede together.

The tractable model in which one wave spreads while the other recedes can give rise to a richer variety of CCFs (Eq. 14). Importantly, the shape of the CCF depends on the direction of the two waves: a reversal in the wave direction reversed the temporal relationship between DA and CINs. Interestingly, the direction of the empirically observed DA waves in dorsal striatum is determined by the nature of the task: medial to lateral waves are associated with instrumental learning, while lateral to medial waves were associated with the reward delivery during classical conditioning[8]. Thus, a prediction of our model would be that each of these tasks should give rise to a different temporal correlation structure. Curiously, the relationship between CINs and midbrain DA neurons does indeed change depending on the nature of the cue, reward, etc.[30,31]. In addition, the ACh waves we report also travel in both directions, although we did not yet establish whether their directionality co-varies with reward contingency. It is therefore possible that while waves seem to occur spontaneously and bidirectionally, they may be biased to occur in particular direction in specific behavioral tasks. In our case, we found a strong inherent bias towards lateral-to-medial waves across experiments, suggesting that lateromedial ACh waves may be associated with behavioral states of inactivity. The fact that brain dynamics exhibit various spontaneous patterns from which a smaller set are selected during sensorimotor behaviors has been described previously[35,98].

The direction of wave propagation required in the model to generate a CCF that resembles empirically observed ones (Fig. 6f, g)[20,32] is such that the CINs wavefront is receding. The delayed peak DA activity relative to CIN activity, actually arises in the model from the drop in CIN activation preceding a drop in DA. This result is actually quite intriguing because the main behaviorally relevant signal of CINs is their famous pause response, in which their firing rate abruptly drops in response to reward or stimuli related to reward[30,40,99–101].

## Turing mechanism

We found that the effects of a Turing bifurcation could be observed both in the regime where the fixed points were high DA/low CIN and vice versa (Fig. 7) and in the regime where the fixed points were high DA/high CIN and low DA/low CIN. In the former case, the fixed point that lost stability was that of high DA/low CIN (e.g., Eq. 7a). In this scenario, while Turing patterns could not be stabilized, the instability could play a role in driving the spontaneous formation of traveling waves. In other words, if the striatum were in a state of low CIN activity, and something drove an elevated global level of DA (e.g., with input from the substantia nigra pas compacta) then that state would lose stability and a traveling wave would form that would drive CIN activity high and lower the DA levels (Fig. 7 and Supplementary Fig. 5). In the latter case, the fixed point that lost stability was that of high DA/high ACh. (e.g., Eq. 18b). In this scenario, a uniform elevation in both CIN and DA would lose stability, and a receding wave of both profiles would form but would leave transient or stable "hills of activity" with the hill of high DA being broader and engulfing the hill of CIN activity (Fig. 10e and Supplementary Figs. 8 and 10). We argued above that the CCFs in this regime give rise to in-phase synchrony between CIN and DA activity which does not agree with current experimental data[20,30–32,97]. Nevertheless, this regime offers a method by which the striatum can undergo a dynamical parcellation or tiling into distinct functional modules of high and low neuromodulatory activity.

The existence of Turing bifurcations and patterns, requires that $D_u/D_v \ll 1$ (Eqs. 25a and b). Is such a regime physiologically relevant? There are two considerations that argue that it may be. First, the axonal arbor of CINs has a radius, $l_{CIN}$, of approximately 0.5 mm[102] whereas the

radius of the axonal arbor of an individual DA, $l_{DA}$, neuron is approximately 1 mm[5,54,103]. Diffusion coefficients scale like the square of their corresponding diffusion lengths. Therefore, we can expect $D_u/D_v < (l_{CIN}/l_{DA})^2$. Physiologically, this means that the axon potential propagation throughout the DA axon arbor will encompass a wider volume than that of the CIN axon arbor. Thus the DA axon arbor will inhibit CINs (via D2Rs) over a longer distance than the excitation of DA axons by CINs (via nAChRs)[21].

## Alternative mechanisms to wave generation

An alternative mechanism to consider is that sequential recruitment of midbrain DA neurons might be the cause of traveling DA waves. Combined with a coherent mapping of the axonal arbors of these neurons in the striatum, sequential activity might give rise to the traveling waves[8,104–106]. Although DA neurons innervate the striatum topographically following a ML course[105,107,108], there is currently no evidence to suggest that midbrain DA neurons are sequentially recruited to support DA waves in the striatum. Also, if the source of the traveling wave is in the midbrain, then the question of mechanism of wave formation just moves one synapse back, and still needs to be explained. It is also questionable whether sequential firing activity at the soma level would reliably translate into waves in the striatum, due to the diverse shapes of DA arbors[5].

A recent in vivo study using fiber photometry from the striatum of mice confirmed the strong temporal correlation between DA and ACh suggested by our model, but did not find strong evidence for local interactions between both modulators locally in the striatum[32,97]. However, because fiber photometry only samples a small brain volume and spatially averages signals within it, this study does not preclude that intrastriatal interactions between DA and ACh contribute to the spreading of waves or to other non-linear spatiotemporal dynamics within the DS. Additional experiments are clearly warranted to reveal the spatiotemporal patterning of DA and ACh release in vivo and to elucidate the underlying mechanisms.

## Summary and central predictions

We have provided evidence for striatal waves of ACh release (Fig. 1), and that nAChRs subserve long range striatal DA release (Fig. 2). Taken alongside the recent observation of striatal waves of DA[8], our findings raise the likelihood that DA and ACh waves are formed by a local (i.e., striatal) mechanism of wave generation. Further work is needed to understand the implication of cholinergic wave-like phenomena for brain dynamics more globally. Several pieces of evidence implicate CINs in behavioral flexibility[109–111]. In particular, it has been suggested that they play a role in ensuring that new learning does not overwrite existing learning[110]. Wave-like spatiotemporal dynamics may provide a mechanism for multiplex operations in the striatal matrix, which could contribute to behavioral flexibility.

We then proposed a physiologically-plausible dynamical mechanism that invokes an AIRD system. Our model makes several predictions that could be tested experimentally. A central prediction to arise from our model is that simultaneous imaging of DA and ACh should reveal waves that are strongly coupled both spatially and temporally, most likely, such that one advances while the other recedes. Another prediction is that blocking nAChRs should compromise the spread of DA waves, and that GIN activity levels control the direction of ACh and DA waves. Finally, enhancing or reducing DA signaling should quench or enhance ACh wave-like behavior. These exciting predictions await experimental confirmation or refutation.

# Methods

## Animals

Experimental procedures adhered to and received prior written approval from the Institutional Animal Care and Use Committees of the Hebrew University of Jerusalem (HUJI, protocol number MD-20-

16113-3), the Okinawa Institute of Science and Technology Graduate University (OIST, protocol number 2021-336-2) and the New York University Grossman School of Medicine (NYUGSOM, protocol number IA16-02082). Most experiments were carried out in C57BL/6J mice (Strain #:000664; Jackson Laboratories, Bar Harbor, ME, USA), with three exceptions. In the iAChSnFR experiments we used ChAT-IRES-Cre (Δneo) transgenic mice (stock number 031661; Jackson Laboratories, Bar Harbor, ME, USA). In the experiments monitoring the activity of CINs in brain slices, we used ChAT-IRES-Cre (Δneo) transgenic mice cross-bred to mice expressing Cre-dependent, Tet-controllable GCaMP6f (Ai148, TIT2L-GC6f-ICL-tTA2; stock number 030328; Jackson Laboratories, Bar Harbor, ME, USA). For experiments in which we studied the release of DA in response to synchronous activation of CINs, we cross-bred the ChAT-IRES-Cre (Δneo) mice with mice expressing Cre-dependent channelrhodopsin-2 [Ai32, RCL-ChR2(H134R)/EYFP; stock number 012569; Jackson Laboratories, Bar Harbor, ME, USA]. Two-to-seven-month-old C57BL/6J and transgenic mice were used for experiments. Sex was not considered as a factor in the design of the experiments, and so mice of both sexes were used and their results pooled. All mice were housed under a 12-h light/dark cycle with food and water ad libitum. Ambient temperatures and humidity were $22 \pm 2$°C and $50 \pm 10$%, respectively.

## Stereotaxic surgeries

At HUJI: Mice were deeply anesthetized with isoflurane in a non-rebreathing system (2.5% induction, 1–1.5% maintenance) and placed in a stereotaxic frame (model 940, Kopf Instruments, Tujunga, CA, USA). Temperature was maintained at 35 °C with a heating pad, artificial tears were applied to prevent corneal drying, and animals were hydrated with a bolus of injectable saline (5 ml/kg) mixed with an analgesic (5 mg/kg carprofen or meloxicam). To calibrate specific injection coordinates, the distance between bregma and lambda bones was measured and stereotaxic placement of the mice was readjusted to achieve maximum accuracy. A small hole was bored into the skull with a micro drill bit and a glass pipette was slowly inserted at the relevant coordinates under aseptic conditions.

A total amount of 250 nl of an adeno-associated virus serotype 5 harboring GRAB-DA2m construct (AAV5-hSyn-DA2m; $> 4.85 \times 10^{13}$ VG/ml viral genome/ml; WZ Biosciences Inc. Lot No. 20210119) was injected with the Nanoject III system (Drummond) into the substantia nigra pars compacta under aseptic conditions. The coordinates of the injection were as follows: anteroposterior, −3.1 mm; mediolateral, +1.2 mm; and dorsoventral, −4.2 mm, relative to bregma using a flat skull position. To minimize backflow, solution was slowly injected and the pipette was left in place for 5–7 min before slowly retracting it.

At NYUGSOM: Mice were implanted with a 3-mm-wide glass window above the entire dorsal surface of the striatum and a head bar as described before[34]. Briefly, mice were administered dexamethasone (4 mg/kg, intraperitoneal) 2 h prior to surgery to minimize brain swelling before being anesthetized with isoflurane (5% induction, 1–1.5% maintenance), placed in a stereotaxic frame (model 940, Kopf Instruments, Tujunga, CA, USA) on a heating blanket (55-7030, Harvard Apparatus) and administered subcutaneously with a bolus of 0.9% sterile saline (1 ml) mixed in with an analgesic (ketoprofen, 10 mg/kg). The scalp was shaved and cleaned with 70% ethanol and iodine scrubs before exposing the skull. A custom titanium headpost was implanted over lambda using clear C&B metabond (Parkell). To achieve wide-spread viral expression of the fluorescent ACh sensor GRAB-ACh3.0[112] throughout the dorsal striatum, 150 nl of AAV9-hSyn-ACh3.0 (Vigene Biosciences; titer $1.46 \times 10^{13}$ VG/ml, diluted 1:2.5 in sterile 0.9% saline) was injected at four distinct locations (anterior/lateral from bregma, in mm): 0.5/1.5, 0.5/2.3, 1.3/1.0, and 1.3/1.8. Injection depth was set at 0.5 mm below the boundary between the dorsal striatum and the external capsule (i.e., 1.7–2.2 mm from dura, depending on injection site). Infusions were carried out at a rate of 100 nl/min using a

microsyringe pump (KD Scientific; Legato 111) fitted with a Hamilton syringe (1701N, gastight 10 µl) connected to a pulled glass injection micropipette (100 µm tip; Drummond Wiretrol II) via PE tubing filled with mineral oil. Injection pipettes were left in place for 5 min before removal. A 3 mm diameter craniotomy was then drilled (centered at 0.9 mm anterior and 1.9 mm lateral from bregma) and cortical tissue was aspirated until the corpus callosum lying above the striatum was exposed. A custom nine gauge thin-walled stainless-steel cannula (Microgroup; 2.3 mm in height) sealed at one end with a 3 mm glass coverslip (Warner Instruments) using optical glue (Norland #71) was placed above the striatum and cemented to the skull using C&B Metabond. Mice were then returned to their home cages and allowed to recover for at least 2 weeks before head-fixation habituation and imaging. Analgesia (ketoprofen, 10 mg/kg) was administered subcutaneously for 3 days after surgery.

At OIST: The genetically encoded fluorescent indicator, iAChSnFR (intensity-based acetylcholine sensing fluorescent reporter)[113,114] was cloned into a partially modified AAV-Tetoff-vector[115]. AAV1-ihSyn-tTA-sv40/TRE-iAChSnFR-minWPRE ($3 \times 10^{13}$ viral genome/ml) was used to express iAChSnFR in DS. In some mice, AAV solution containing the static red indicator, AAV1-CAG-DIO-tdTomato ($4 \times 10^{13}$ viral genome/ml) was also injected.

All surgical procedures were conducted under aseptic conditions. Mice were anesthetized with isoflurane (3% induction, 1–2% maintenance) and placed in a stereotaxic frame (Kopf Instruments). A local anesthetic, lidocaine was applied at the incision site, after which a small incision was performed along the midline using a sterile scalpel to expose the skull. To achieve expression of iAChSnFR, mice were injected unilaterally at target coordinates with either AAV1-ihSyn-tTA-sv40/TRE-iAChSnFR-minWPRE alone or in combination with AAV1-CAG-DIO-tdTomato. The coordinates of the injection were as follows: AP, +0.5 mm, +1.0 mm; ML, +1.5 mm, +2.0 mm; and DV −2.8 mm, relative to bregma. The injection was done through a glass pipette using a nanoliter injector (Nanoject III, Drummond). The glass pipette containing AAV solution was slowly moved down to the target coordinates over the course of a few minutes. The AAV solution was slowly injected (200 nl per injection site), and the pipette was left in place for 5–10 min to allow diffusion after which it was slowly retracted to minimize backflow. Following viral injection, a metal head-plate was affixed to the skull with adhesive cement (C&B, Sun Medical) and a self-curing dental acrylic resin (Unifast II, GC Corporation). Mice were then returned to their home cages and allowed to recover. Analgesia (carprofen for post-operative treatment, dissolved in 0.9% saline, 5 mg/kg, i.p.) was administered for 3 days after surgery.

One week after virus injection and headplate installation, the mice were instrumented with either a GRIN lens or a cranial window. A 1 mm diameter GRIN lens (Inscopix, Palo Alto, CA, USA) was implanted into the dorsal striatum. All surgical procedures were performed under aseptic conditions as above. To facilitate the insertion of the GRIN lens, a 25 G needle was first inserted from the brain surface to a depth of about 200 µm above the GRIN lens target position. After removal of the needle, the GRIN lens was then lowered to approximately 200 µm directly above the injection site. The lens was fixed in place using Kwik sil (WPI) and dental acrylic resin. Mice were monitored over a recovery period of 2–3 weeks after which they were habituated to head restraint before imaging commenced.

Alternatively, cranial window implant surgery was performed 1 week after virus injection. Mice were again anaesthetized and placed in the stereotaxic frame. Dexamethasone (dissolved in 0.9% saline, 2 mg/kg, s.c.) was administered to reduce brain edema. A circular craniotomy (approximately 2.5 mm diameter), centered around the virus injection coordinates was performed on the skull overlying the dorsal striatum. The dura was removed, and the overlying cortex was slowly aspirated using a blunted 25 G needle connected to a vacuum pump. A circular ring with a round coverglass (Warner Instruments) adhered to one side was slowly inserted into the craniotomy and gently placed over the striatal surface. The implant was fixed to the skull using dental cement. Mice were then returned to their cages and allowed to recover. Carprofen was administered for 3 days after surgery.

## Widefield in vivo imaging

At NYUGSOM: Following recovery from surgery in their home cage, mice were habituated to handling by the experimenter and to being head restrained while on a cylindrical treadmill[116] placed in a dark soundproof chamber for a minimum of 5 days prior to imaging. Widefield epifluorescence imaging of the entire dorsal aspect of the striatum was performed through a 4X long working-distance air objective (Olympus XLFLUOR4X-340) mounted on a Thorlabs Bergamo-II microscope with epifluorescence module (WFA2001) and camera port (WFA4101). Full-field ACh3.0 fluorescence was imaged from the right hemisphere continuously (20 Hz frame rate, 50 ms exposure, 696 × 510 pixels after 2 × 2 binning) for a minimum of 10 min using an X-Cite 110LED illuminator, a green fluorescent protein (GFP) filter cube (Thorlabs TLV-U-MF2-GFP; Excitation: $469 \pm 18$ nm, Emission: $525 \pm 20$ nm) and a monochrome CCD camera (Thorlabs 1500M-GE).

At OIST: Widefield imaging was conducted approximately 3 weeks following GRIN lens or cranial window implantation. Images were acquired from the right hemisphere using a Retiga Electro CCD camera (Teledyne QImaging) with an Objective lens (×5 or ×10, Zeiss), 20 Hz frame rate and ×4 on-camera binning. iAChSnFR was excited with 470 nm LED (Thorlabs) and detected using ET520/40 emission filter (Thorlabs). To reduce light contamination, the imaging chamber was covered with a black-out curtain. Time-series movies of spontaneous ACh signals were acquired during periods of inactivity while head-restrained mice remained stationary in a dark imaging chamber, and without external stimuli. Imaging sessions consisted of 3–5 approximately 1-min-long recordings.

## Acute slice preparation

Two to three weeks after the viral injections mice were deeply anesthetized with ketamine (200 mg/kg)–xylazine (23.32 mg/kg) and perfused transcardially with ice-cold modified artificial cerebrospinal fluid (ACSF) bubbled with 95% $O_2$–5% $CO_2$, and containing (in mM): 2.5 KCl, 26 $NaHCO_3$, 1.25 $Na_2HPO_4$, 0.5 $CaCl_2$, 10 $MgSO_4$, 0.4 ascorbic acid, 10 glucose, and 210 sucrose. The brain was removed and sagittal slices sectioned at a thickness of 275 µm were obtained in ice-cold modified ACSF. Slices were then submerged in ACSF, bubbled with 95% $O_2$–5% $CO_2$, and containing (in mM): 2.5 KCl, 126 NaCl, 26 $NaHCO_3$, 1.25 $Na_2HPO_4$, 2 $CaCl_2$, 2 $MgSO_4$, and 10 glucose, and stored at room temperature for at least 1 h prior to recording.

## Slice visualization, 2PLSM imaging, and electrophysiology

The slices were transferred to the recording chamber of Femto2D-resonant scanner multiphoton system (Femtonics Ltd., Budapest, Hungary) and perfused with oxygenated ACSF at 32 °C. A ×16, 0.8 NA water immersion objective was used to examine the slice using oblique illumination.

2PLSM $Ca^{2+}$/monoamine imaging: The 2PLSM excitation source was a Chameleon Vision 2 tunable pulsed laser system (680–1080 nm; Coherent Laser Group, Santa Clara, CA). Optical imaging of GCaMP6f/GRAB-DA2m signals was performed by using a 920-nm excitation beam. The fluorescence emission was detected and collected with gated GaAsP photomultiplier tubes (PMTs) for high sensitivity detection of fluorescence photons as part of the Femto2D-resonant scanner. Areas of approximately 340 µm × 340 µm were selected and imaged with 100 µm intervals from the electric stimulation center at 31 Hz scans were performed, using 0.6 µm pixels. Regions-of-interest were marked manually offline. The system is also equipped with full-field 470 nm LED illumination, which was used to stimulate the CINs in the

ChAT-ChR2 mice. The gated PMTs are turned off for a time window of a few tens of milliseconds flanking the LED pulse. Nevertheless, there is a light artifact in the signal from the PMTs that get saturated by the LED pulse even in the absence of an active voltage.

For whole-cell current-clamp recordings from CINs, the pipette contained (in mM): 135.5 $KCH_3SO_4$, 5 KCl, 2.5 NaCl, 5 Na-phospho-creatine, 10 HEPES, 0.2 EGTA, 0.21 $Na_2GTP$, and 2 $Mg_{1.5}ATP$ (pH = 7.3 with KOH, 280–290 mOsm/kg). Alexa Fluor 568 (30 µM, Invitrogen) were added to the patch pipette to visualize the CIN morphology. CINs were identified by their morphology, spontaneous firing and by the presence of a voltage-sag (due to their prominent HCN currents). CINs were hyperpolarized to quiescence (−10 to −150 pA) prior to injection of depolarizing currents, as follows: (a) for bursts:150 pA, 200 ms; (b) for single spikes: 2 nA, 0.5 ms. A square region of 133 µm × 133 µm around the patched CIN was selected and imaged at approximately 31 Hz scans were performed, using 0.26 µm pixels. Regions-of-interest were marked manually offline based on online fluorescence response.

Optical and electrophysiological data were obtained using the software package MESc (Femtonics), which also integrates the control of all hardware units in the microscope. The software automates and synchronizes the imaging signals and electrophysiological protocols. Electrophysiological recordings were obtained with a Multiclamp 700B amplifier (Molecular Devices, Sunnyvale, CA). Signals were digitized at 10–40 kHz and logged onto a personal computer with the MESc software (Femtonics). Data was extracted from the MESc package (Femtonics) to personal computers using custom-made code in MATLAB (MathWorks, Natick, MA, USA) code. Fluorescent changes over time ($\Delta F/F_0$) datapoints were extracted such that $\Delta F/F_0 \stackrel{\text{def}}{=} \frac{F-F_0}{F_0}$, where $F$ is the maximal fluorescent value recorded while evoking electrical stimulation; $F_0$ denotes the averaged baseline fluorescence.

### Electrical stimulation protocol
Electrical stimulation was carried out by a parallel bipolar Platinum-Iridium electrode with diameter of 125 µm and spacing of 500 µm (FHC, PBSA0575). The magnitude of the stimulus was controlled through a stimulus isolator (ISO-Flex, MicroProbes) while the frequency and duration were controlled by computer software (MESc, Femtonics). A total of 10 pulses (10 Hz, 2 ms duration, 3 mA) were delivered.

### Imaging and statistical analysis
**2PLSM resonant scanning.** In Fig. 2c, $\Delta F/F_0$ data points were fitted with a Lorentzian function $\frac{a}{1+(x/\lambda)^2} + c$ to extract the length scale $\lambda$ of the fluorescence signals' decay with distance. Individual mice were considered independent samples. Due to the nested design of this experiment, wherein we sampled multiple slices from individual mice, we fit the data with a linear mixed-effects model (LMEM), where $\Delta F/F_0$ is modeled as a product of two fixed effect, which are the treatment (e.g., mecamylamine vs. control) and the distance, plus uncorrelated random effects due to each individual mouse. To fit the LMEMs to the data, we used the Matlab (Mathworks) FITLME command. We reported the number of slices, the number of mice, the absolute value of Student's $t_v$ statistic (where $v$ is the number of degrees of freedom) and the two-tailed $P$ value of the fixed effects. The null hypotheses was rejected if the $P$ value was below 0.05.

### Pre-processing of ACh imaging movies.
Fluorescent movies sampled at 20 frames per second were motion corrected with the motion-correction module of MIN1PIPE algorithm[117] (Because of their large size the GRAB-ACh movies were then spatially down-sampled using ImageJ). The fluorescent signals were z-scored for further analysis, such that $Z = \frac{F-\bar{F}}{\sigma}$, where $F$ denotes the raw fluorescence recorded, $\sigma_F$ is the mean standard deviation across approximately 1.5 second-long periods (with 0.75 second-long overlaps) throughout the measurement, and $\bar{F}$ is the mean overall fluorescence. Similarly, fluorescence changes

over time ($\Delta F/F_0$) were extracted for visualization, such that $\Delta F/F_0 = \frac{F-F_0}{F_0}$, where $F$ represents the raw fluorescence recorded and $F_0$ denotes the minimal averaged fluorescence.

**Space–time representation of waves and estimation of wave location.** In each video we visually determined a custom-shaped freehand region-of-interest (ROI) that fit the area in which changes in the signal were most significant. We then clustered the pixels in the ROI to form bands perpendicular to the absolute ML axis (Fig. 1b). In each frame we performed spatial averaging by calculating the mean z-scored signal across each perpendicular band, thus reducing it to a single point in space at a single point in time. This space–time representation of the data enabled us to visualize the dynamics of the signal (Fig. 1c, d). Furthermore, at each point in time we calculated the position (band) where the z-scored signal was maximal, a point approximating the location of the wave. In order to test whether this location is governed by slow temporal dynamics, that potentially arise artificially from the slow, filtering properties of ACh indicators (GRAB-ACh3.0 or iAChSnFR)[112,118], we performed a statistical analysis using the boot-strapping method.

**Bootstrapping.** First we calculated the temporal derivative of the location of the maximal activity, which gives an estimate of the instantaneous velocity of the wave (Fig. 1d, bottom). Then, the vector of the locations of the maximal activity was permuted, and the resulting velocity vector was re-calculated for each permutation. For each permutation, we measured the mean duration of runs between sign reversals of the velocity. This permutation was conducted 1000 times and maximal mean spurious duration were extracted. Because the fluorescent signal is expected to exhibit temporal correlations (due to the slow kinetics of the indicators), we repeated this process by changing the nature of the permutations: we divided the vector of locations into contiguous chunks of length $n$, where $n$ varied between 1 (50 ms) and 20 (1000 ms). We then permuted these chunks before calculating the duration between the resulting velocity reversal. As $n$ is increased from 1 to 20 the maximal mean spurious duration between velocity reversals increased.

**Detection of individual wave events, calculation of their duration, frequency, and velocity.** We developed an algorithm to detect wave events in the space-time rendition. First, the algorithm identified sequences of frames (runs, at least 5 frames long) in which the velocity vector maintains a mediolateral direction in the majority of the frames in the run. This first step identified candidate time windows in the video when the trajectory of the location of the maximal fluorescence advanced mediolaterally. Trajectories that were either too short or their average slope in the mediolateral direction was too shallow were discarded. The remaining candidate runs were divided into at least 3 sub-sequences (depending on the length of the run). In order to pro-nounce a run as a wave, we first calculated the average z-scored signal within the entire ROI of each of the sub-sequences. Next, for all pairs of adjacent sub-sequences of the run, we calculated the difference in their average signal intensity, and determined whether this difference was smaller than 70% of the mean intensity of the first of these two sub-sequences. If somewhere along the entire run of length $R$, there was a consecutive series of length >$0.4R$ of adjacent pairs that fulfilled this condition, then the whole run was deemed a wave of length $R$, pro-vided one additional condition was fulfilled: the average intensity of the signal in the medial region of the first frame of the entire candidate run was at least 2.5 the average signal intensity in the lateral region (this last condition ensured that the wave began with activity that was indeed restricted to the medial aspect of the ROI). It is evident that there are many free parameters in this algorithm, and indeed they were manually fine-tuned per movie so that they reliably identified what looked upon visual inspection as mediolateral diagonal streaks in the

space-time rendition. This process was then repeated for the later-omedial direction. The same parameters were used for mediolateral and lateromedial wave detection.

For the events identified as waves, we calculated the inter-wave interval by subtracting the start times of adjacent waves, and the wave interval as the length $R$ of the wave. In order to calculate the mean velocity of a wave event we averaged the value of the instantaneous velocity vector corresponding to the frames that were part of that wave.

### Turing instability

In our models, we consider situations where the presence of diffusion will cause one of the uniform fixed points of Eqs. 6a and b to become destabilized through a Turing bifurcation to form Turing patterns[70]. These are spatial activity patterns of a particular spatial scale that is determined by the parameters of the problem. For this analysis, we will need to consider the stability matrix, $A$, of Eqs. 6a and b, which will have the form

$$A_{(u^*,v^*)} = \begin{pmatrix} \frac{\partial f_i(u^*)}{\partial u} & -1 \\ \frac{\partial g_i(u^*)}{\partial u} & -1 \end{pmatrix} \tag{24}$$

Where $(u^*,v^*)$ is one of the stable fixed point of Eqs. 6a and b in the absence of diffusion. The conditions for the uniform solution $(u^*,v^*)$ to lose stability through a Turing bifurcation are: (a) $A_{11} > 0$; (b) $A_{12}A_{21} < 0$; (c) trace of $A$ is negative; and (d)

$$D_u/D_v < \frac{1}{A_{22}^2}\left(|A| - A_{12}A_{21} - 2\sqrt{-A_{12}A_{21}|A|}\right) \tag{25a}$$

where $|A|$ is the determinant of $A$[70]. Because $A_{12} = A_{22} = -1$, Eq. 25a simplifies to

$$D_u/D_v < |A| + A_{21} - 2\sqrt{A_{21}|A|} \tag{25b}$$

In the models that we analyzed, calculations show that: the trace of $A$ (i.e., $\frac{\partial f_i(u^*)}{\partial u} - 1$) will be negative; $|A| = \frac{\partial g_i(u^*)}{\partial u} - \frac{\partial f_i(u^*)}{\partial u}$ will be positive; and $A_{21} = \frac{\partial g_i(u^*)}{\partial u}$ will be positive in the entire parameter regime of the candidate fixed-point $(u^*,v^*)$ (the other stable fixed point cannot undergo a Turing bifurcation). Geometrically, this means that at the stable fixed point $(u^*,v^*)$, that can lose stability through a Turing instability, the slope of $g_i(u)$ is always greater than the slope of $f_i(u)$, (that latter of which is always smaller than 1). Thus, by the inequality of arithmetic and geometric means, the right-hand-side of Eqs. 25a and b is always positive, so there will always be a small enough ratio $D_u/D_v$ to attain a Turing bifurcation, provided $A_{11} > 0$. Thus, in the models we will analyze, the parameter regime where Turing bifurcations are attainable is defined by the curve $\frac{\partial f_i(u^*)}{\partial u} > 0$ (which is indicated as yellow regions in the phase diagrams in Figs. 7c and 10c). Geometrically, what this boils down to, is that the slope of $f_i(u)$ at the Turing bifurcation point is positive, which is why $f_i(u)$ must have an inverted-N shape. Physiologically, this means that CINs must excite themselves, which we proposed could happen through recurrent-disinhibition, due to the putative U-shaped dependence on CIN activity of the nAChRs (located on the GINs).

### Reporting summary

Further information on research design is available in the Nature Portfolio Reporting Summary linked to this article.

### Data availability

The raw data generated in this study belonging to Figs. 1–3 and Supplementary Figs. 1 and 2 have been deposited in Dryad https://doi.org/10.5061/dryad.b5mkkwhk8. Source data are provided with this paper.

### Code availability

All the information necessary to run the XPPAUT simulations is present in the paper, and the codes themselves (including parameter, initial conditions and settings) are provided in the Supplementary Code.

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

## Acknowledgements

This work was funded by a Research Grant from the Human Frontier Science Program (HFSP) no. RGP0062/2019 to J.R.W. and J.A.G.; by an European Research Council (ERC) Consolidator Grant (no. 646886) to J.A.G.; and by grants from the National Institutes of Health (DP2NS105553 and R01MH130658) and Dana and Whitehall Foundations to N.X.T. We thank Loren Looger for providing iAChSnFR and Kiyoto Kurima for cloning iAChSnFR into a partially modified AAV-Tetoff-vector.

## Author contributions

Conceptualization, N.X.T., J.R.W and J.A.G.; methodology, L.M., N.G., G.A.S., Y.A., L.T., N.X.T. and J.A.G.; software, L.M., N.G., Y.A., L.T. and J.A.G.; validation, L.M., N.G., G.A.S., Y.A., L.T. and J.A.G.; analysis, L.M., N.G., G.A.S., Y.A., L.T. and J.A.G.; resources, J.A.G.; data curation, L.M., N.G., G.A.S., Y.A., L.T., N.X.T. and J.A.G.; writing, L.M., N.G., G.A.S., Y.A, J.R.W. and J.A.G.; visualization, L.M., N.G. and J.A.G.; supervision, J.A.G.; project administration and funding acquisition, N.X.T., J.R.W. and J.A.G.

## Competing interests

The authors declare no competing interests.
