## [Peer Review File · Nature Communications]

Acetylcholine waves and dopamine release in the striatumReviewers' Comments:

Reviewer #1:

Remarks to the Author:

Matityahu et al study wave-like phenomena in the striatum. They build upon the recently discovered spatiotemporal waves of dopamine (DA) release to propose that the cholinergic neuropil (CIN) is also activated in a wave like fashion. They hypothesized that the DA and the CIN wave-like dynamics are coupled. They provide some evidence of this coupling in a slice experiment. Specifically they show that bipolar stimulation elicits a bump in activity of DA neurons and that the spatial decay constant is decreased by pharmacologic blockade of the AChR. Based on these observations, they propose a model formulated as a reaction diffusion mixture. A key ingredient of this model is the observation that the activation function of AChR is an inverted U-shape. They do a formal bifurcation analysis of the reaction-diffusion system to explore the parameters that give rise to the wave-like phenomena and make a variety of predictions concerning the correlation between DA and CIN. These predictions await experimental confirmation.

Overall comments: Overall, I found the modeling component of the work compelling and interesting. The authors have done a thorough job of analyzing the equations by appropriately simplifying them and then performing a rigorous bifurcation analysis. I also agree with their formulation of the system as a reaction diffusion.

That being said, the fact that wave-like phenomena can arise in reaction diffusion systems is well known as is acknowledged by the authors. Thus, the specific interest in this particular instantiation of a reaction diffusion system relies critically on how well it can explain experimental observations. This correspondence between theory and experiment, however, is not well explored in the present manuscript. Given the fact that, albeit imperfect, methods for doing this exist, it will really strengthen the paper if some of the predictions of the model were demonstrated experimentally by simultaneously imaging DA and CIN wave-like activity. Furthermore, clarification of the existing experimental observations would really strengthen the argument.

Major Comments

1. Figure 1 and the accompanying movie form the bulk of the experimental observations that are meant to demonstrate the existence of waves in CIN neuropil. Specifically, Figure 1D is supposed to show the percolation of the wave. It is not clear (at least to me) how these data show the existence of the wave? I have a really hard time evaluating this claim on the basis of the raw signal. It seems that the authors assert the fact that the wave exists on the basis of the optic flow algorithm. Yet, it is not clear to me how well the direction of motion of the activity bump inferred by the optic flow algorithm actually captures the experimental data. The goodness of fit of the optical flow model should be demonstrated.

2. Furthermore, it appears that the optical flow was applied to just the upswings (but not the downswings) of the activity. It is not clear why this is justified. The methodology of how exactly the data were selected for the optic flow calculation is not clearly explained (lines 732-741). It is not clear to me why 85% cutoff was used. It is also not clear why, if a random frame was selected (not necessarily at the beginning of the upswing), is there a temporal averaging over L frames (where L is the average duration of upswing in units of frames). Why is temporal average necessary in the first place? The following statement is unclear: "We considered the vectors calculated for the upswing as significant if its modulus exceeded the threshold associated with its pixel". How is the threshold computed? What statistical test was used? Analysis of these data should really benefit from comparison to a null model where by construction waves do not occur.

3. The trace of the spatially averaged activity of CIN neuropil makes it appear that the activity is periodic in time. Specifically, it appears that the activity trace can be well represented by addition of two slightly out of phase sine waves. Yet, this temporal periodicity is not explained or commented on.

Perhaps this is not a reliable feature of the data? In any case this deserves some consideration.

4. One really good bit of news for the model is that, in the absence of performing simultaneous recordings of both CIN and DA activity, the model under some parameter choices can give rise to cross correlation function (CCF) similar to that observed in Howe et al (eLife 2019), present manuscript Figures 5F-G. This is a really nice find. However, Howe et al go on to show that both the DA and CIN activity change dramatically as a function of the movement state of the animal and can either converge or diverge depending on the movement state. It is surprising that although the movement state of the animal was not taken into account, the authors find highly reproducible spatial gradient vectors across upswings and across days. This deserves a comment. Also, can the proposed model account for the movement-dependent combined dynamics suggested by Howe et al?

5. A critical ingredient of the model that gives rise to the wave-like phenomena is the inverted U-shaped curve of AChR. As the authors appropriately discuss, this is an in vitro result that awaits in vivo justification. However, given the importance of this aspect of the model, it would have been good to see how the decrease in the spatial decay constant (slice experiments in Figure 2) depends on the concentration of the AChR antagonist.

Reviewer #2:

Remarks to the Author:

This manuscript presents data and a mathematical model on the relationship between DA and ACh signals in the mouse striatum. The paper builds on a recent body of work that has identified wave-like behavior of striatal DA. The data in this manuscript reveal some coordinated activity of ACh neurons that is consistent with waves in these neurons, and the slice experiments presented in the manuscript are suggestive for ACh-mediated enhancement of striatal DA signaling. The model posits that ACh neurons are activators and DA neurons are inhibitors, and that the reciprocal interactions between these neurons, together with inhibitory neurons, might generate interlinked wave-like patterns of these two modulatory transmitter systems in the striatum.

Fundamentally, the manuscript addresses an interesting and important problem (interactions between DA and ACh in the striatum), and current data in the literature are strongly suggestive (albeit debated) for DA waves. However, the manuscript falls short in several important ways, which in my view precludes publication at this point.

Main concern:

The proposed model builds on data that do not establish ACh wave-like behavior, and the presented work does not test key predictions of the new model experimentally. In general, while the data in the literature on DA waves are quite strong, the data on ACh waves presented here are not. The ACh wave-like phenomenon builds on a previous paper (refs 25 and 26, a biorxiv version that strongly advocates for wave-like behavior and a published version of the same manuscript that is more descriptive). The data from this previous paper are re-analyzed in Fig. 1 of the current manuscript, but overall, the data and analyses are so limited that ACh waves are not established. At most, these data are consistent with wave-like behavior. The paper then goes on to present a limited dataset in brain slices (Fig. 2), in which GRAB-DA signals evoked by electrical stimulation are more widespread than GCAMP signals in ACh neurons. This suggests that DA expands beyond ACh and is consistent with recent work on this topic. However, brain slices are not suited to study wave-like behavior of DA, because in the slices, the DA axons are cut off from their somata, and hence they lack key ascending activity patterns (tonic and phasic firing). Furthermore, outcomes of cross-comparisons of different sensors (GRAB-DA and GCAMP) may be strongly influenced by sensor properties, which is not addressed at all. The rest of the paper then proposes and tunes a mathematical model. This model makes key predictions on the behavior of DA and ACh in the striatum, but the manuscript does not go back and experimentally test these predictions. What stays, in the end, is a model that is neither

rooted in strong experimental data, nor are new predictions tested with experiments. As such, the work appears premature.

Additional specific points:

- The manuscript, the way it is written, does not distinguish well between assumptions and outcomes of the model. The authors should present a section upfront on the assumptions that were made to generate the model.
- Wave-like behavior of ACh would have to be established more rigorously before building a model.
- One prediction of the model is that blocking ACh receptors should reduce DA waves. This could be tested in vivo.
- Another prediction is that enhancing or reducing DA would quench or enhance ACh wave-like behavior (if present), respectively, this could be tested in vivo,
- Another prediction is that individual CINs should induce local DA release. This is inconsistent with much of the literature on the topic (refs. 18-20, 22, 29 and the literature that has led to these relatively recent studies). It is also improbable because the activation threshold for axonal sodium channels is likely below the threshold of axonal calcium channels (which are presumably needed for DA release). Without direct experimental evidence, the impact of the model is limited in this respect.
- Two recent papers (refs. 18, 22) propose that ACh induces spiking in DA axons to mediate release, at least in brain slices. It appears that much of the model is built around local activation ($D_v = 0$), and the induction of spiking is hardly discussed and not well incorporated in the model, apart from brief section in the end. Propagating voltage in the DA axon should be better incorporated in the work here.
- The wording around the information provided by modeling should be clarified. This is best illustrated by the last sentence of the abstract ("Our model provides a biophysical mechanism for wave formation..."). The model does not establish a biophysical mechanism. Instead, it can provide a descriptive framework for how an experimentally established biological process may work, and/or make predictions on the presence of new mechanisms that can then be tested experimentally. The paper falls short on both points.

Reviewer #3:

Remarks to the Author:

The current manuscript provides a very important confirmation of DA waves. DA waves are interesting phenomena that were observed and need investigation. Also, it demonstrates CHAT waves happen at the same time. Also manuscript suggests a very sophisticated computational model that I really enjoy. Obviously, a) common input mechanisms and b) interaction between interneuron networks (SOM, CHAT and PV) can produce the same results. Finally, waves can be formed via the DA-Striatum-> DA loop. I strongly suggest incorporating additional models with other possible scenarios. Finally, by feeding data in multiple plausible models authors may suggest which model is more biologically possible. General comments 1) current manuscript can not focus only on one type of interneuron that performs a "DA editing". There is multiple evidence from March Fuccillo lab (UPENN) that also SOM interneurons control a dopamine release in a similar way as CHAT interneurons. So that needs to be incorporated into it the model or at least discussed. 2) Finally, all interneurons have complex interactions. Therefore in order to claim selectivity, multiple agonist and antagonist need to be used. Or at least take it into consideration in the model. 3) "activator inhibitor reaction diffusion" assumption is not necessary. However a loop like activation, DA-> interneuron (s)-> DA is better assumption 4) Also maybe more realistic is striatum->DA-Striatum loop with a spatial shift in the loop 5) Second "global role" of the waves in striatal computation needs to be investigated. Maybe that is a "chain" of computation similar to synfire chain? Some chain of neuronal cluster activations was observed in the cortex using MEG. Maybe authors want to consult with Moshe Abeles to brainstorm a global computational role.

RESPONSE REVIEWER COMMENTS

Reviewer #1 (Remarks to the Author):

Reviewer Comment:

Overall comments: Overall, I found the modeling component of the work compelling and interesting. The authors have done a thorough job of analyzing the equations by appropriately simplifying them and then performing a rigorous bifurcation analysis. I also agree with their formulation of the system as a reaction diffusion. That being said, the fact that wave-like phenomena can arise in reaction diffusion systems is well known as is acknowledged by the authors. Thus, the specific interest in this particular instantiation of a reaction diffusion system relies critically on how well it can explain experimental observations. This correspondence between theory and experiment, however, is not well explored in the present manuscript. Given the fact that, albeit imperfect, methods for doing this exist, it will really strengthen the paper if some of the predictions of the model were demonstrated experimentally by simultaneously imaging DA and CIN wave-like activity. Furthermore, clarification of the existing experimental observations would really strengthen the argument.

Author Response:

We very much appreciate the reviewer's comment that they found the modelling compelling and interesting. We agree that the correspondence between theory and experiment was not explored well in the original submission. We have therefore undertaken additional experiments and modelling. In the revised manuscript we included additional results bearing on the model. These include two independently obtained data sets in which acetylcholine waves were directly imaged in awake animals using two different genetically encoded sensors for acetylcholine in two different labs. These results and the analysis of wave-like properties are reported in the revised manuscript as follows (page 4, lines 97-123):

We conducted fluorescence imaging of a genetically-encoded ACh sensor (GRAB-ACh3.0) expressed in the dorsal striatum (DS) of 2 head-fixed mice via a 3 mm diameter cranial window³⁵ (Fig. 1a). Visualization of the ACh signal demonstrated spatiotemporal patterns similar to those exhibited by DA in the DS⁸, as the ACh signal could be visualized traveling across the DS (Suppl. Movie 1) primarily along the ML axis. In order to analyze the wave activity, we averaged the activity perpendicular to the ML axis (Fig. 1b), and tracked it over time. Diagonal streaks in the space-time rendition of this activity demonstrated the occurrence of waves that move along the ML axis (Fig. 1c,d). Strikingly, tracking the location of the peak activity in space (Fig. 1c,d, dots) showed that location of the peak activity changed gradually in time with an instantaneous ML velocity that fluctuated in the ± 10 mm/s range (Fig. 1d). Bootstrapping demonstrated that the motion of the peak activity of the ACh signal is inconsistent with random spatial activations³⁶ (Fig. 1e). We extracted wave events with a heuristic algorithm (see Methods), and estimated the distributions of wave durations and inter-wave intervals (Fig. 1f). From these distributions, we could estimate that waves occurred on average once every 5.4 ± 0.6 seconds (mean \pm sem), and that their mean duration was 394 ± 11 ms. Importantly, the inter-wave intervals distributed across multiple time scales ranging from sub-second to 10s of seconds, demonstrating that they occurred irregularly. Using the velocity curves (Fig. 1d, bottom), we extracted the distribution of the mean velocity of the waves (Fig. 1g). Interestingly, approximately 80% of the waves spread from lateral-to-medial. Similar results were obtained by imaging another genetically-encoded ACh sensor (iAChSnFR) expressed in the DS via a cranial window with a 1 mm diameter field-of-view (N =2 mice) or via a 1 mm diameter GRIN lens (N =1 mouse, Fig. 1h, Suppl. Movie 2). Here too, the ACh signal formed waves that exhibited a strong preference to travel in the lateral-to-medial direction (Fig. 1i). In these

mice, ACh waves occurred on average every 7 ± 0.6 seconds, and their mean duration was 591 ± 43 ms (Fig. 1j).

We agree that it would strengthen the paper to simultaneously image DA and CIN wave-like activity but we have been unable to do this primarily because of technical limitations related to the spatial period of the acetylcholine waves. The acetylcholine imaging revealed a wavelength on the order of a millimeter, which requires a somewhat wider field of view to adequately visualise. The field of view available for dual color imaging in our in vivo set-ups turned out to be insufficient. Our two-photon setup is capable of multichannel imaging but has an even narrower field of view. On the other hand, although we can image on this scale in epifluorescence, our epifluorescence set-up only allows single color imaging. Since DA waves are relatively well established compared to acetylcholine waves, we directed our effort during the revisions toward imaging ACh waves.

Reviewer Comment:

Major Comments

1. Figure 1 and the accompanying movie form the bulk of the experimental observations that are meant to demonstrate the existence of waves in CIN neuropil. Specifically, Figure 1D is supposed to show the percolation of the wave. It is not clear (at least to me) how these data show the existence of the wave? I have a really hard time evaluating this claim on the basis of the raw signal. It seems that the authors assert the fact that the wave exists on the basis of the optic flow algorithm. Yet, it is not clear to me how well the direction of motion of the activity bump inferred by the optic flow algorithm actually captures the experimental data. The goodness of fit of the optical flow model should be demonstrated.

2. Furthermore, it appears that the optical flow was applied to just the upswings (but not the downswings) of the activity. It is not clear why this is justified. The methodology of how exactly the data were selected for the optic flow calculation is not clearly explained (lines 732-741). It is not clear to me why 85% cutoff was used. It is also not clear why, if a random frame was selected (not necessarily at the beginning of the upswing), is there a temporal averaging over L frames (where L is the average duration of upswing in units of frames). Why is temporal average necessary in the first place? The following statement is unclear: "We considered the vectors calculated for the upswing as significant if its modulus exceeded the threshold associated with its pixel". How is the threshold computed? What statistical test was used? Analysis of these data should really benefit from comparison to a null model where by construction waves do not occur.

Author Response:

We agree with the reviewer's comment that these observations and the analysis were unclear in the first submission. We have taken these observations out of the paper for several reasons in addition to the referee's comments, including concerns about the small number of mice precluding statistical analysis and a lack of data on the cardinal directions of the wave travel. Instead of relying on the data from CIN neuropil, we conducted additional experiments using acetylcholine sensors. We also had an opportunity to use a data set that had been independently collected by Nic Tritsch who had done these experiments and seen more convincing evidence of waves. Thus we have added two new data sets (collected by Dr. Tritsch in NYU and by Prof. Wickens' lab in OIST), that are based on two different sensors and both show convincing evidence of acetylcholine waves, as described in the new Fig. 1 and accompanying text.

We have also adopted a new method of analysis and applied it to each data set. Because the DA waves were shown to travel mostly in the mediolateral aspect (Hamid et al. Cell 2020), and because visual inspection of our data (see supplementary movies) demonstrated that this is largely the case with respect to the ACh waves, we simplified the analysis by collapsing the ACh signals to this mediolateral (ML) aspect (by averaging all the pixels perpendicular to a point on the

ML aspect of that point). This allowed us to generate an intuitive space–time diagram, where space (the signal projected onto the ML aspect) is along the y-axis and time is along the x-axis. In this rendition it became clear that there were diagonally oriented streaks of activity which represent ACh release that progresses along the ML aspect, and became our definition of a wave (new Fig. 1). To demonstrate statistically that the spatiotemporal trajectory of maximal activity was significant we applied a bootstrapping method to show that the signal does not fluctuate randomly in space. These results and the analysis of wave-like properties are reported in the revised manuscript as detailed in the preceding response (above).

Reviewer Comment:

3. The trace of the spatially averaged activity of CIN neuropil makes it appear that the activity is periodic in time. Specifically, it appears that the activity trace can be well represented by addition of two slightly out of phase sine waves. Yet, this temporal periodicity is not explained or commented on. Perhaps this is not a reliable feature of the data? In any case this deserves some consideration.

Author Response

We agree that this possibility deserves some consideration, even though the comment refers to the original neuropil data set, which we have replaced with the acetylcholine concentration data sets. Insofar as the frequency of wave occurrence is concerned, it is irregular as can be seen in the inter-wave intervals that are distributed across multiple time scales ranging from sub-second to 10s of seconds (New Figs. 1f,j). Therefore, there is no periodicity in the occurrences of waves. We now describe these findings as follows (page 4, lines 109-115):

We extracted wave events with a heuristic algorithm (see Methods), and estimated the distributions of wave durations and inter-wave intervals (Fig. 1f). From these distributions, we could estimate that waves occurred on average once every 5.4 ± 0.6 seconds (mean \pm sem), and that their mean duration was 394 ± 11 ms. Importantly, the inter-wave intervals distributed across multiple time scales ranging from sub-second to 10s of seconds, demonstrating that they occurred irregularly.

Having said that, the trajectory of the maximal ACh activity along the ML axis seems to have some regularity in time (new Fig. 1d). To test for this we measured the spectra of this trajectory. Indeed, while the spectrum of the trajectory show a peak in the 1Hz, this was true only for one of the two mice expressing GRAB-ACh3.0 and imaged with a 3 mm diameter cranial window (see panel A in the appended figure). However, none of the mice expressing iAChSnFR imaged with either a 1mm diameter GRIN lens or a 1 mm diameter cranial windows exhibit an periodicity (see panel B). So, we have no strong indication of periodicity in the spread of ACh release in DS.

The spectrum of the trajectory of maximal fluorescence along the mediolateral aspect shows a weak 1 Hz periodicity in one mouse expressing GRAB-ACh3.0 (black trace in panel A). In contrast, the spectrum of the trajectory from the other mouse did not (gray trace in panel A). The trajectories extracted from the other 3 mice that expressed iAChSnFR and were imaged via a 1 mm aperture exhibited a “ $1/f$ ” spectrum with no periodicity (panel B).

Reviewer Comment:

4. One really good bit of news for the model is that, in the absence of performing simultaneous recordings of both CIN and DA activity, the model under some parameter choices can give rise to cross correlation function(CCF) similar to that observed in Howe et al (eLife 2019), present manuscript

Figures 5F-G. This is a really nice find. However, Howe et al go on to show that both the DA and CIN activity change dramatically as a function of the movement state of the animal and can either converge or diverge depending on the movement state. It is surprising that although the movement state of the animal was not taken into account, the authors find highly reproducible spatial gradient vectors across upswings and across days. This deserves a comment. Also, can the proposed model account for the movement-dependent combined dynamics suggested by Howe et al?

Author Response:

In the experimental acetylcholine dataset the wave activity was not movement-dependent. The time-series of ACh signals were acquired while the head-restrained mice remained stationary in a dark chamber, without external stimuli. However, unlike in the neuropil data, we found that the waves in the ACh signals could travel in both directions (although there was a marked bias towards lateromedial over mediolateral progression). Thus, it is possible that when the mouse is not engaged in a task there are nevertheless spontaneous waves, and that when the animal engages in a certain behavior there is an associated preferential direction, as was observed by Hamid et al. (2020) regarding DA waves. We mention this possibility in the Discussion as follows (page 20, lines 578-586):

In addition, the ACh waves we report also travel in both directions, although we did not yet establish whether their directionality co-varies with reward contingency. It is therefore possible that while waves seem to occur spontaneously and bidirectionally, they may be biased to occur in particular direction in specific behavioral tasks. In our case, we found a strong inherent bias towards lateral-to-medial waves across experiments, suggesting that lateromedial ACh waves may be associated with behavioral states of inactivity.

To address this question in the model, we demonstrate a property of the model that we did not delve into previously. We now show that changes in the basal activity of the interneuronal populations represented in the model – for example a sudden increase in the feedforward activation of the GABAergic interneurons – can cause a reversal in the direction of the wave. Thus, there exists in the model a mechanism by which the direction of the wave can change on a moment-by-moment basis, which would in turn alter the structure of the cross-correlation function. This is described in the Results section on page 10, lines 293-6:

The parameter $A = I_C - \beta I_G$ (Eqs. 4 & 5a) can also control the direction of the wave propagation (Suppl. Fig 3b), indicating that global changes in activity levels of (or common inputs to) the interneurons (see Eqs. 1 & 2) can affect the reciprocal dynamics between CINs and DA axons.

And then again in the Discussion on page 17 lines 475-482:

Interestingly, the full model includes another parameter, A , (Eq. 6a) that can alter the direction of motion. A reduction in A shifts $f_1(u)$, the u -nullcline (e.g., Fig. 4a), downwards, which alters the areas between the nullclines that determine the direction of the wave. $A = I_C - \beta I_G$ (Eqs. 4 & 5a) represents a linear combination of the autonomous (and/or synaptic) drive to CINs (Eq. 1) and GINs (Eq. 2). Thus, if I_G is increased, A is reduced, which can, in turn, change the direction of the wave (Suppl. Fig. 3b). Thus, our model does enable changes in the drive to the various striatal interneurons to alter the direction of the wave propagation on a moment-by-moment basis.

Reviewer Comment:

5. A critical ingredient of the model that gives rise to the wave-like phenomena is the inverted U-shaped curve of AChR. As the authors appropriately discuss, this is an *in vitro* result that awaits *in vivo* justification. However, given the importance of this aspect of the model, it would have been

good to see how the decrease in the spatial decay constant (slice experiments in Figure 2) depends on the concentration of the AChR antagonist.

Author Response

We appreciate the comment and agree that it would have been good to see how different concentrations of the antagonist modify the spatial decay constant. However, we do not expect the method we used to be sufficiently sensitive to produce statistically significant differences in degree depending on antagonist concentration. We usually use antagonists in sufficient doses for an all-or-none effect. An experiment to determine a graded effect would require a very large number of replicates over a wide range of concentrations, given the effect size and variability measured with 10 μ M mecamylamine, and this was not feasible in the available time frame.

Reviewer #2 (Remarks to the Author):

Reviewer Comment:

This manuscript presents data and a mathematical model on the relationship between DA and ACh signals in the mouse striatum. The paper builds on a recent body of work that has identified wave-like behavior of striatal DA. The data in this manuscript reveal some coordinated activity of ACh neurons that is consistent with waves in these neurons, and the slice experiments presented in the manuscript are suggestive for ACh-mediated enhancement of striatal DA signaling. The model posits that ACh neurons are activators and DA neurons are inhibitors, and that the reciprocal interactions between these neurons, together with inhibitory neurons, might generate interlinked wave-like patterns of these two modulatory transmitter systems in the striatum.

Fundamentally, the manuscript addresses an interesting and important problem (interactions between DA and ACh in the striatum), and current data in the literature are strongly suggestive (albeit debated) for DA waves. However, the manuscript falls short in several important ways, which in my view precludes publication at this point.

Main concern:

The proposed model builds on data that do not establish ACh wave-like behavior, and the presented work does not test key predictions of the new model experimentally. In general, while the data in the literature on DA waves are quite strong, the data on ACh waves presented here are not. The ACh wave-like phenomenon builds on a previous paper (refs 25 and 26, a biorxiv version that strongly advocates for wave-like behavior and a published version of the same manuscript that is more descriptive). The data from this previous paper are re-analyzed in Fig. 1 of the current manuscript, but overall, the data and analyses are so limited that ACh waves are not established. At most, these data are consistent with wave-like behavior.

Author response:

As noted above in our response to a similar concern raised by Reviewer #1 (see Major Comments) we have performed additional experiments and obtained additional data, and applied a new analysis approach that strengthens the evidence for acetylcholine waves. We have now observed ACh waves using fluorescence imaging of genetically-encoded ACh sensors, GRAB-ACh3.0, and iAChSnFR, in the dorsal striatum of head-fixed mice, and report these results as follows (page 4, lines 97-123):

We conducted fluorescence imaging of a genetically-encoded ACh sensor (GRAB-ACh3.0) expressed in the dorsal striatum (DS) of 2 head-fixed mice via a 3 mm diameter cranial window³⁵ (Fig. 1a). Visualization of the ACh signal demonstrated spatiotemporal patterns similar to those exhibited by DA in the DS⁸, as the ACh signal could be visualized traveling across the DS (Suppl. Movie 1) primarily

along the ML axis. In order to analyze the wave activity, we averaged the activity perpendicular to the ML axis (Fig. 1b), and tracked it over time. Diagonal streaks in the space-time rendition of this activity demonstrated the occurrence of waves that move along the ML axis (Fig. 1c,d). Strikingly, tracking the location of the peak activity in space (Fig. 1c,d, dots) showed that location of the peak activity changed gradually in time with an instantaneous ML velocity that fluctuated in the ± 10 mm/s range (Fig. 1d). Bootstrapping demonstrated that the motion of the peak activity of the ACh signal is inconsistent with random spatial activations³⁶ (Fig. 1e). We extracted wave events with a heuristic algorithm (see Methods), and estimated the distributions of wave durations and inter-wave intervals (Fig. 1f). From these distributions, we could estimate that waves occurred on average once every 5.4 ± 0.6 seconds (mean \pm sem), and that their mean duration was 394 ± 11 ms. Importantly, the inter-wave intervals distributed across multiple time scales ranging from sub-second to 10s of seconds, demonstrating that they occurred irregularly. Using the velocity curves (Fig. 1d, bottom), we extracted the distribution of the mean velocity of the waves (Fig. 1g). Interestingly, approximately 80% of the waves spread from lateral-to-medial. Similar results were obtained by imaging another genetically-encoded ACh sensor (iAChSnFR) expressed in the DS via a cranial window with a 1 mm diameter field-of-view (N =2 mice) or via a 1 mm diameter GRIN lens (N =1 mouse, Fig. 1h, Suppl. Movie 2). Here too, the ACh signal formed waves that exhibited a strong preference to travel in the lateral-to-medial direction (Fig. 1i). In these mice, ACh waves occurred on average every 7 ± 0.6 seconds, and their mean duration was 591 ± 43 ms (Fig. 1j).

Reviewer Comment:

The paper then goes on to present a limited dataset in brain slices (Fig. 2), in which GRAB-DA signals evoked by electrical stimulation are more widespread than GCAMP signals in ACh neurons. This suggests that DA expands beyond ACh and is consistent with recent work on this topic. However, brain slices are not suited to study wave-like behavior of DA, because in the slices, the DA axons are cut off from their somata, and hence they lack key ascending activity patterns (tonic and phasic firing). Furthermore, outcomes of cross-comparisons of different sensors (GRAB-DA and GCAMP) may be strongly influenced by sensor properties, which is not addressed at all.

Author Response

As noted in the previous comment, wave-like behaviour was studied in the intact brain. Brain slices were used specifically to study the potential of direct interaction between CINs and DA axons under conditions in which the DA axons were detached from the soma. This approach has been utilized in several influential studies that showed the direct effect of CIN stimulation on DA release (Threlfell et al. 2012; Cachope et al. 2012; Liu et al. Science 2022; and others). We agree that it is important to take into account the properties of sensors when comparing the spatial decay of activation along the axons of CINs with the spatial decay of DA release from DA axons. Two points mitigate this concern. First, the important comparison is the spatial decay of DA release in slices in control conditions with the spatial decay in the presence of the nAChR antagonist, mecamylamine. As this is a within-slice experimental design (i.e. the same conditions with and without mecamylamine), the same sensor properties operate in both the control and experimental conditions. Second, for the determination of the spatial extent of recruitment of CINs, the use of a different sensor is unavoidable. However, parameters in which various fluorophores differ primarily concern their temporal properties. The experiment we conducted has to do with spatial decay, which should not be strongly affected by any of these properties. We comment about this final point in the text as follows (page 5, lines 146-169):

While differences in sensor properties could theoretically affect the comparison of spatial scales of GCaMP6f and GRAB-DA2m, the sensors differ primarily in their temporal properties,^{38, 39, 40} which should not strongly affect spatial decay.

Reviewer Comment:

The rest of the paper then proposes and tunes a mathematical model. This model makes key predictions on the behavior of DA and ACh in the striatum, but the manuscript does not go back and experimentally test these predictions. What stays, in the end, is a model that is neither rooted in strong experimental data, nor are new predictions tested with experiments. As such, the work appears premature.

Author Response

We understand the reviewer's concern and have extended the experimental testing of predictions of the model as follows:

The main prediction of the model is the existence of acetylcholine waves. This prediction stemmed from our previous observation of waves in the neuropil and the model that it directed us to. This prediction has now been tested in additional experiments in which we independently measured acetylcholine waves directly *in vivo* using two different genetically encoded sensors for acetylcholine (new Fig 1). This is the first report of acetylcholine travelling waves, validating this prediction of the model with new data, as detailed in a preceding response (above)

Another important prediction of the model is that intrastriatal dynamics are sufficient to generate acetylcholine and dopamine travelling waves, and that sequential external input is not necessary. The new data added to the paper was obtained from head-fixed animals in a dark space that were not engaged in a task. This shows that acetylcholine travelling waves are not movement-related or caused by sensory stimulation and occur even when the animal is in a state of quiet rest.

An important assumption of the model is that individual cholinergic interneurons should induce local dopamine release (without requiring synchronous activation of several CINs – because synchrony among CINs is not an ingredient of the model, instead individual CINs should be able to induce local DA release). We have now performed additional experiments and provide new data showing that individual cholinergic interneurons can induce local dopamine release in striatum (Fig 3).

Reviewer Comment:

Additional specific points:

- The manuscript, the way it is written, does not distinguish well between assumptions and outcomes of the model. The authors should present a section upfront on the assumptions that were made to generate the model.

Author Response

In the revised manuscript an overview of the assumptions that were made to generate the model has been added, as follows (page 7, line 205-215):

An overview of the assumptions made in the construction of the model is as follows. To capture the activator-inhibitor relation we assume that CINs activate nAChRs on DA axons to increase DA release, and conversely that DA activates D2Rs on CINs to decrease ACh release. We assume that activation of nAChRs has an inverted-U shaped dependence on concentration of ACh with fast kinetics. To capture the reaction-diffusion interaction we assume that both DA and ACh axonal arbors can be represented by a spatially extended variable, they fill the space and interact at all points, and the spread of activity is governed by the cable properties of the axons. The CINs are assumed to be self-exciting (or rather self-disinhibiting), by receiving inhibition from GABA

interneurons (GINs) that flip the inverted-U shaped nonlinearity of nAChR activation into an inverted-N shaped dependence of CIN activation on ACh concentration.

In addition a section summarizing the outcomes of the model has been added at the end of the Results section, as follows (page 15, lines 407-421):

Summary of model predictions

Several testable predictions can be derived from the model. In addition to predicting the existence of ACh traveling waves intertwined with DA waves, the model predicts that the local interaction between DA axons and CINs is sufficient to generate waves. External input is not necessary, suggesting that traveling waves can occur even when the animal is in a state of quiet rest, as was observed both with respect to DA⁸ and ACh (Fig. 1) waves. Two other predictions are that under conditions of no external input, DA-ACh interaction, as formulated in the model, is necessary to generate waves, and that APs in individual CINs cause DA release from axons (without requiring CIN synchrony). Conversely, blocking ACh receptors should reduce both DA and ACh waves. Similarly, enhancing or reducing DA signaling would alter DA and ACh wave-like behavior. The model also predicts that the direction of the wave motion can be reversed by a sudden increase in the feedforward activation of the GINs. Under certain conditions isolated hills of activity can occur (Appendix 2). Although we have validated the first few of these predictions, further experimental work would be required to verify the latter predictions of the model.

And briefly again at the end of the Discussion, as follows (pages 22-23, lines 654-660):

Our model makes several predictions that could be tested experimentally. A central prediction to arise from our model is that simultaneous imaging of DA and ACh should reveal waves that are strongly coupled both spatially and temporally, most likely, such that one advances while the other recedes. Another prediction is that blocking nAChRs should compromise the spread of DA waves, and that GIN activity levels control the direction of ACh and DA waves. Finally, enhancing or reducing DA signaling should quench or enhance ACh wave-like behavior.

Referee Comment:

- *Wave-like behavior of ACh would have to be established more rigorously before building a model.*

Author Response

As noted above in our response to a similar concern raised by Reviewer #1 (see Major Comments) we have now observed ACh waves using fluorescence imaging of genetically-encoded ACh sensors, GRAB-ACh3.0, and iAChSnFR, in the dorsal striatum of head-fixed mice, and report these results as follows (page 4, lines 97-123):

We conducted fluorescence imaging of a genetically-encoded ACh sensor (GRAB-ACh3.0) expressed in the dorsal striatum (DS) of 2 head-fixed mice via a 3 mm diameter cranial window³⁵ (Fig. 1a). Visualization of the ACh signal demonstrated spatiotemporal patterns similar to those exhibited by DA in the DS⁸, as the ACh signal could be visualized traveling across the DS (Suppl. Movie 1) primarily along the ML axis. In order to analyze the wave activity, we averaged the activity perpendicular to the ML axis (Fig. 1b), and tracked it over time. Diagonal streaks in the space-time rendition of this activity demonstrated the occurrence of waves that move along the ML axis (Fig. 1c,d). Strikingly, tracking the location of the peak activity in space (Fig. 1c,d, dots) showed that location of the peak activity changed gradually in time with an instantaneous ML velocity that fluctuated in the ± 10 mm/s range (Fig. 1d). Bootstrapping demonstrated that the motion of the peak activity of the ACh signal is inconsistent with random spatial activations³⁶ (Fig. 1e). We extracted wave events with a heuristic algorithm (see Methods), and estimated the distributions of wave durations and inter-

wave intervals (Fig. 1f). From these distributions, we could estimate that waves occurred on average once every 5.4 ± 0.6 seconds (mean \pm sem), and that their mean duration was 394 ± 11 ms. Importantly, the inter-wave intervals distributed across multiple time scales ranging from sub-second to 10s of seconds, demonstrating that they occurred irregularly. Using the velocity curves (Fig. 1d, bottom), we extracted the distribution of the mean velocity of the waves (Fig. 1g). Interestingly, approximately 80% of the waves spread from lateral-to-medial. Similar results were obtained by imaging another genetically-encoded ACh sensor (iAChSnFR) expressed in the DS via a cranial window with a 1 mm diameter field-of-view (N =2 mice) or via a 1 mm diameter GRIN lens (N =1 mouse, Fig. 1h, Suppl. Movie 2). Here too, the ACh signal formed waves that exhibited a strong preference to travel in the lateral-to-medial direction (Fig. 1i). In these mice, ACh waves occurred on average every 7 ± 0.6 seconds, and their mean duration was 591 ± 43 ms (Fig. 1j).

Reviewer Comment:

- One prediction of the model is that blocking ACh receptors should reduce DA waves. This could be tested *in vivo*.
- Another prediction is that enhancing or reducing DA would quench or enhance ACh wave-like behavior (if present), respectively, this could be tested *in vivo*.

Author Response

We agree that these tests of model predictions suggested by the referee would be great experiments and are the way to go in the future. Regrettably, we were not able to do these tests in the timeframe of the revisions, in which we focussed on increasing confidence in the existence of acetylcholine travelling waves, and experimentally testing a key assumption of the model. At this stage we have included the reviewer's suggestions as predictions of the model and added these to the summary of model predictions quoted in a previous response (above).

Reviewer Comment:

- Another prediction is that individual CINs should induce local DA release. This is inconsistent with much of the literature on the topic (refs. 18-20, 22, 29 and the literature that has led to these relatively recent studies). It is also improbable because the activation threshold for axonal sodium channels is likely below the threshold of axonal calcium channels (which are presumably needed for DA release). Without direct experimental evidence, the impact of the model is limited in this respect.

Author Response

We have performed additional experiments and provide new data showing that individual cholinergic interneurons can induce local dopamine release in striatum. In these experiments GRAB-DA2m was injected into the substantia nigra pars compacta and used to measure dopamine release in acute striatal slices while evoking action potential bursts from a patched cholinergic interneuron. Increases in dopamine were observed in several regions of interest distant from the soma and in particular around dopamine axons. These data are reported in Fig 3 and described in the Results section on pages 5-6 , lines 158-174:

Because the prevailing view is that only synchronous activation of multiple CINs can induce localized DA release^{21, 22, 23, 26}, we combined two-photon laser scanning microscopy (2PLSM) imaging of GRAB-DA2m, expressed selectively in dopaminergic axons, with patch-clamp recordings from individual CINs in acute striatal slices (Fig. 3a). Indeed, we found in 21% of the patched CINs ($n = 6/29$ CINs from $N = 6/13$ mice) that evoking a burst of action potentials (APs) can cause a measurable increase in DA in the vicinity the CIN. Measurements in various regions-of-interest localized at different areas of the axonal arbor demonstrate that the signal is visible throughout the CIN's arbor (Fig. 3b), suggesting that a single CIN can influence DA release as far as its axonal field extends. In some cases,

we were able to locate an individual stretch of axon that exhibits an even larger amplitude of release, presumably because the DA concentration is highest near the releasing axon that expressed the DA sensor (Fig. 3b, red region-of-interest). In one CIN, we were even able to observe DA release in response to a single AP (Fig. 3c). Using this method, DA release could also be observed in response to rebound spiking after hyperpolarizing the CIN, which may be important in the context of the pause response exhibited by CINs^{31, 41, 42}.

Reviewer Comment:

- Two recent papers (refs. 18, 22) propose that ACh induces spiking in DA axons to mediate release, at least in brain slices. It appears that much of the model is built around local activation ($D_v = 0$), and the induction of spiking is hardly discussed and not well incorporated in the model, apart from brief section in the end. Propagating voltage in the DA axon should be better incorporated in the work here.

Author Response

We regret that we did not make this topic clearer, and we hope we remedied it now. Using $D_v = 0$ made several aspects of the model analytically-tractable, which in our humble opinion, is a strong point of the current study. We also noted that making D_v non-zero, which indeed corresponds to the spreading of the activation throughout the putative DA syncytium, did not qualitatively change the findings regarding traveling waves. Moreover, the entire discussion of Turing instability requires D_v to be non-zero. We have therefore tried to make these points more explicit in the revised manuscript. First on page 9, lines 261-269:

We will consider two regimes in our analysis. In the first we will assume that $D_v = 0$, which corresponds to a regime where activation of nAChRs on DA fibers, can induce local activation of these fibers (and presumably release of DA), but does not cause (electrical) activity to propagate throughout the DA axonal arbor. This regime is amenable to analysis. In the second regime D_v is allowed to be non-zero, which corresponds to electrical activity propagating throughout DA axonal arbor, as was recently demonstrated^{21, 26}. We shall see that $D_v > 1$ does not qualitatively alter the traveling wave solutions. However, it can give rise under certain conditions to the appearance of Turing patterns³⁴ of isolated hills of activity.

Then again on page 14, lines 379-381.

Thus, whether D_v is zero or non-zero does not qualitatively alter the traveling wave solutions, indicating that striatal waves can occur whether or not nAChRs trigger a local traveling AP in DA axons, as was recently shown^{21, 26}.

Reviewer Comment:

The wording around the information provided by modeling should be clarified. This is best illustrated by the last sentence of the abstract (“Our model provides a biophysical mechanism for wave formation...”). The model does not establish a biophysical mechanism. Instead, it can provide a descriptive framework for how an experimentally established biological process may work, and/or make predictions on the presence of new mechanisms that can then be tested experimentally. The paper falls short on both points.

Author Response

While we understand why the use of the term “biophysical” seems inappropriate to the reviewer, in our humble opinion, the use of the theory of dynamical systems to explain biological phenomena is an acceptable and some would say elegant way to explain collective phenomena such as spatially distributed and coincident neuronal activity or neuromodulator release. We

believe it also has strong explanatory value, and therefore provides much more than merely describing the phenomena. Moreover, our model does make falsifiable predictions which we now list more explicitly at the end of Results and at the end of the Discussion (as described in a previous response).

We also want to reiterate that finding striatal ACh waves was in and of itself a prediction that stemmed out of this study (both from our previous observation of waves in the neuropil and the model that it directed us to). This prediction was borne out between the original submission and the current revision of this manuscript. We therefore respectfully disagree with the reviewer's opinion that the paper falls short on either of its explanatory or its predictive power. Nevertheless we have re-worded the final sentence of the Abstract as follows (page 1, lines 30-32):

Thus, our study provides evidence for striatal acetylcholine waves *in vivo*, and proposes a testable theoretical framework that predicts that the observed dopamine and acetylcholine waves are strongly coupled phenomena.

Reviewer #3 (Remarks to the Author):

Reviewer Comment

The current manuscript provides a very important confirmation of DA waves. DA waves are interesting phenomena that were observed and need investigation. Also, it demonstrates CHAT waves happen at the same time. Also manuscript suggests a very sophisticated computational model that I really enjoy. Obviously, a) common input mechanisms and b) interaction between interneuron networks (SOM, CHAT and PV) can produce the same results. Finally, waves can be formed via the DA-Striatum-> DA loop. I strongly suggest incorporating additional models with other possible scenarios. Finally, by feeding data in multiple plausible models authors may suggest which model is more biologically possible.

General comments

- 1) current manuscript can not focus only on one type of interneuron that performs a "DA editing". There is multiple evidence from March Fuccillo lab (UPENN) that also SOM interneurons control a dopamine release in a similar way as CHAT interneurons. So that needs to be incorporated into it the model or at least discussed.*
- 2) Finally, all interneurons have complex interactions. Therefore in order to claim selectivity, multiple agonist and antagonist need to be used. Or at least take it into consideration in the model*

Author Response

We appreciate the reviewer's interest in the phenomena observed and enthusiasm for the model. While we certainly agree that the model is not the only one possible to explain dopamine waves, we focussed our revisions on increasing confidence in the existence of acetylcholine waves, the putative mechanism for generating them, and providing evidence of the interaction between individual cholinergic interneurons and dopamine release (as detailed in responses to reviewers above). The action of SOM interneurons (which we refer to as LTSIs) is mediated by direct action on GABA receptors, as shown by the Fuccillo lab (Holly et al, 2020; Holley et al, 2021). This action is inhibitory in contrast to the excitatory effect of CINs, and as such cannot produce waves by the activator inhibitor reaction diffusion mechanism we have proposed. They showed that this modulation does not depend on acetylcholine, by performing their experiments while blocking the effects of acetylcholine. [They wrote, "Potential contributions of ChINs were eliminated by optogenetically stimulating dopamine terminals expressing Cre-dependent channelrhodopsin and performing all experiments in the presence of antagonists for cholinergic nicotinic $\beta 2$ (DH β E) and muscarinic (scopolamine) receptors. ChINs predominately regulate dopamine via nicotinic $\beta 2$

subunits, and $\beta 2$ antagonism disrupts ChIN-dopamine interactions to a similar degree as broad nicotinic antagonists such as mecamylamine.” Holly et al 2021, page 4139.]

The action of GABA interneurons *are* included in the “full model” with $f_1(u)$ and $g_1(u)$ in the parameter A , which can also control the direction of wave propagation, as shown in Suppl. Fig 3B. To address the reviewer comment we have added some discussion of the evidence that LTSIs control dopamine release and how this impacts the model as follows (page 17, lines 475-488).

Interestingly, the full model includes another parameter, A , (Eq. 6a) that can alter the direction of motion. A reduction in A shifts $f_1(u)$, the u -nullcline (e.g., Fig. 4a), downwards, which alters the areas between the nullclines that determine the direction of the wave. $A = I_C - \beta I_G$ (Eqs. 4 & 5a) represents a linear combination of the autonomous (and/or synaptic) drive to CINs (Eq. 1) and GINs (Eq. 2). Thus, if I_G is increased, A is reduced, which can, in turn, change the direction of the wave (Suppl. Fig. 3b). Thus, our model does enable changes in the drive to the various striatal interneurons to alter the direction of the wave propagation on a moment-by-moment basis. Interestingly, a recent study has shown that LTSIs can attenuate DA release in the striatum⁶⁷. Because their action is inhibitory LTSIs are not likely to drive DA waves but they may be able to influence them. Indeed, in our model, it is precisely GINs, for whom $I_G > 0$, that can control the direction of wave propagation. Because $I_G > 0$ represents autonomously active GINs, such as LTSIs,^{57, 58, 59} our model actually predicts that the activity of LTSIs should be able to affect the direction of ACh as well as DA waves in the striatum.

Reviewer Comment

3) "activator inhibitor reaction diffusion" assumption is not necessary. However a loop like activation, DA-> interneuron (s)-> DA is better assumption

Author Response

If we understand correctly, the reviewer suggests an alternative way to realize an “auto-catalytic” population in the model. Instead of our proposal of self-excitation of CINs by ACh, it is suggested that a di-synaptic feedback loop of DA axons onto themselves via GABA interneurons could attain the same self-disinhibitory effect. This is an interesting suggestion. However, in that scenario the CINs would need to be the inhibitory population, and have a larger “diffusion coefficient”. Neither of these are consistent with the morphology and the effect of CINs on DA axons. We have discussed other potential mechanisms that could give rise to self-excitation, on page 18, lines 519-530:

However, it is possible that alternative mechanisms of self-excitation exist *in vivo* that could support our model and give rise to traveling waves. For example, DA fibers can co-release glutamate^{50, 70, 71, 72, 73}. Activation of nAChRs on DA might lead to glutamate release, which in turn would both excite CINs on a fast time scale through ionotropic glutamate receptors that would effectively provide self-excitation, and inhibit them more slowly by activation of metabotropic D2Rs⁵⁰. Alternatively, activation of presynaptic $\alpha 7$ nAChRs on cortical terminals might produce self-excitation of CINs⁶⁸. In such a scenario, activation of cortical input could introduce the bi-stability: when $\alpha 7$ nAChRs were activated, CINs would become autocatalytic because in this scenario their release of ACh could cause the cortical input to have a stronger excitatory influence over themselves causing self-excitation. This alternative scenario has the corollary that in the absence of cortical drive CINs cannot be autocatalytic.

Reviewer Comment

4) Also maybe more realistic is striatum->DA-.Striatum loop with a spatial shift in the loop

Author Response

We agree that the waves could be inherited from afferent inputs to the striatum (e.g., from the substantia nigra pars compacta). However, we argue that then we would need to understand what generated the wave in the afferent input, which might be due to a similar or different mechanism. We also agree that the spiralling structure of the reciprocal connections between the substantia nigra and the striatum (Haber) could generate reverberating activity with a spatial shift. However, it is not immediately obvious how the direction of the resulting wave could reverse so frequently. In order to be more even handed, we have touched upon some of the alternative mechanisms, and have quoted findings from a new study that challenge the idea of local coupling between ACh and Dain the striatum (pages 21-22, lines 623-641):

Alternative mechanisms to wave generation

An alternative mechanism to consider is that sequential recruitment of midbrain DA neurons might be the cause of traveling DA waves. Combined with a coherent mapping of the axonal arbors of these neurons in the striatum, sequential activity might give rise to the traveling waves^{8, 88, 89, 90}.

Although DA neurons innervate the striatum topographically following a ML course^{89, 91, 92}, there is currently no evidence to suggest that midbrain DA neurons are sequentially recruited to support DA waves in the striatum. Also, if the source of the traveling wave is in the midbrain, then the question of mechanism of wave formation just moves one synapse back, and still needs to be explained. It is also questionable whether sequential firing activity at the soma level would reliably translate into waves in the striatum, due to the diverse shapes of DA arbors⁵.

A recent *in vivo* study using fiber photometry from the striatum of mice confirmed the strong temporal correlation between DA and ACh suggested by our model, but did not find strong evidence for local interactions between both modulators locally in the striatum³³. However, because fiber photometry only samples a small brain volume and spatially averages signals within it, this study does not preclude that intrastriatal interactions between DA and ACh contribute to the spreading of waves or to other non-linear spatiotemporal dynamics within the DS. Additional experiments are clearly warranted to reveal the spatiotemporal patterning of DA and ACh release *in vivo* and to elucidate the underlying mechanisms.

Reviewer Comment

5) Second "global role" of the waves in striatal computation needs to be investigated. Maybe that is a "chain" of computation similar to synfire chain? Some chain of neuronal cluster activations was observed in the cortex using MEG. Maybe authors want to consult with Moshe Abeles to brainstorm a global computational role.

Author Response

We agree with the referee that cholinergic waves in the striatum are likely to have effects on cortical dynamics, and in particular with the synfire chains proposed by Moshe Abeles. However, in truth our study focuses more on the dynamics than the function. We have added the following text to the Discussion about possible roles for the ACh waves (Page 22, Line 646-652):

Further work is needed to understand the implication of cholinergic wave-like phenomena for brain dynamics more globally. Several pieces of evidence implicate CINs in behavioral flexibility^{93, 94, 95}. In particular, it has been suggested that they play a role in ensuring that new learning does not overwrite existing learning⁹⁴. Wave-like spatiotemporal dynamics may provide a mechanism for multiplex operations in the striatal matrix, which could contribute to behavioral flexibility.

Reviewers' Comments:

Reviewer #2:

Remarks to the Author:

The authors have made significant adjustments in response to the previous round of review. I particularly appreciate (1) the discussion of the assumptions of the model that makes it a much more accessible resource and (2) the addition of new in vivo data that are at least indicative for ACh waves in the mouse striatum.

In my view, several limitations persist, in particular in the connection of the model to biological data.

- Fig 1 is a significant addition, but the way the data are analyzed it is not clear that wave-like behavior of striatal ACh activity is a robust phenomenon that is reproduced across animals and experiments. A measure that would allow cross-comparison of animals, and probably inclusion of more animals for a given method, would strengthen the analyses.
- Fig 3 is also a new addition. For 6 neurons (CINs), out of 29 neurons from a total of 13 mice, some correlation of activation of a single CIN leading to an increase in GRAB-DA fluorescence is observed. The data in the figure contain selected traces from 4 neurons/imaging fields. Each example uses a different stimulation protocol, it is unclear whether within an image area the phenomenon is repeatable, and no attempt for establishing causality (for example a before-after experiment with nAChR blockers) is made. The authors conclude, quite strongly, that "activation of a single CIN suffices to induces local striatal DA release" (lines 81-82) and "...overturning the widely-held view that synchronous activation of several CINs is required" (lines 431-432). While this would be a very important conclusion, it is not scientific to use the presented data to make this conclusion so strongly. As it stands, it is a correlation in a few example traces without an attempt to establish causality and reproducibility, and for most neurons it appears not to be the case. Systematic experimentation and analyses would be required. At best, the current data suggest that rarely, a single CIN might suffice to induce dopamine release, but in most cases it might not. At this point, the data are circumstantial.
- Model assumption: the model assumes that DA and ACh axonal arbors interact at all points. This is not the case. Direct connections are very sparse (PMID: 8684624), and the striatum has a high concentration of acetylcholine esterase and inhibiting it affects both the frequency and kinetics of ACh-to-DA transmission (PMID: 35931070). Clearly, their interactions are limited.
- The simplest and most straightforward prediction of the model is that inhibiting ACh receptors should impair dopamine waves. Without that test, much of the model remains unvalidated.
- Clear descriptions of the number of observations and statistics in the figure legends are missing.
- In several figures (for example 1f-1j), y-axis labels are missing.

Altogether, there remain significant weaknesses in the biological data and in the assessment of their relationship to the model. The existence of ACh waves and their influence on DA waves remains uncertain.

Reviewer #3:

Remarks to the Author:

Understanding DA wave mechanisms is critical for the understanding of striatal function and dysfunction. This manuscript suggests a possible mechanism underlying this wave's existence. My suggestions were focused on alternative mechanisms of these waves. These mechanisms were discussed and modeled in the revised manuscript. I hope this work will inspire new experimental work to prove or improve the model. In my view, this work is ready to be presented to the public.

RESPONSE TO REVIEWER COMMENTS

Reviewer #1

No additional comments received.

Reviewer #2

Reviewer comment:

The authors have made significant adjustments in response to the previous round of review. I particularly appreciate (1) the discussion of the assumptions of the model that makes it a much more accessible resource and (2) the addition of new in vivo data that are at least indicative for ACh waves in the mouse striatum. In my view, several limitations persist, in particular in the connection of the model to biological data.

-Fig 1 is a significant addition, but the way the data are analyzed it is not clear that wave-like behavior of striatal ACh activity is a robust phenomenon that is reproduced across animals and experiments. A measure that would allow cross-comparison of animals, and probably inclusion of more animals for a given method, would strengthen the analyses.

Author response

We thank the Reviewer for their careful re-evaluation of our manuscript. Our revised manuscript now contains additional analyses and data to address most of the Reviewer's outstanding concerns. We hope the Reviewer will find our study improved and worthy of publication. We address specific points in the sections below:

Regarding the robustness of the ACh wave activity, we added data from mice imaged with GRAB-ACh3.0 (now N = 3) and iAChSnFR (now N = 6) reaching a total of N = 9 mice, which is more than the number of mice reported in the original description of DA waves. We also now provide data on three measures of the wave statistics: their duration, velocity and inter-wave intervals, and show that these measure are consistent across mice and ACh sensor (see **Fig. 1g,i** and the **new Suppl. Fig. 1 a,b**). We believe these speak strongly to the robustness of the wave-like behavior of striatal ACh activity and its reproducibility across animals (using two different indicators in two separate labs). These data are reported as follows in our revised manuscript (lines 102-130, in the clean copy):

In order to analyze the wave activity, we averaged the activity perpendicular to the ML axis (Fig. 1b), and tracked it over time. Diagonal streaks in the space-time rendition of this activity demonstrated the occurrence of waves that move along the ML axis (Fig. 1c,d). Strikingly, tracking the location of the peak activity in space (Fig. 1c,d, dots) showed that location of the peak activity changed gradually in time with an instantaneous ML velocity that fluctuated in the ± 10 mm/s range (Fig. 1d).

Bootstrapping demonstrated that the motion of the peak activity of the ACh signal is inconsistent with random spatial activations³⁶ (Fig. 1e). We extracted wave events with a heuristic algorithm (see Methods), and estimated the distributions of wave durations and inter-wave intervals (Fig. 1f), **which were consistent across mice (Suppl. Fig. 1a)**. From these distributions, we could estimate that waves occurred on average once every 5.2 ± 0.5 seconds (mean \pm sem), and that their mean duration was 391 ± 9 ms. Importantly, the inter-wave intervals distributed across multiple time scales ranging from sub-second to 10s of seconds, demonstrating that they occurred irregularly. Using the velocity curves (Fig. 1d, bottom), we extracted the distribution of the mean velocity of the waves, **which were consistent across mice (Fig. 1g)**. Interestingly, approximately 80% of the waves spread from lateral-to-medial.

We obtained similar results by imaging another genetically-encoded ACh sensor (iAChSnFR) expressed in the DS via a cranial window with a 1 mm diameter field-of-view (N =2 mice) or via a 1 mm diameter GRIN lens (N =4 mice, Fig. 1h, Suppl. Movie 2). Here too, the ACh signal formed waves that exhibited a strong preference for travel in the lateral-to-medial direction (Fig. 1i) **consistently**

across mice (Suppl. Fig. 1b,c). In these mice, ACh waves occurred on average every 8.6 ± 1.2 seconds, and their mean duration was 537 ± 18 ms (Fig. 1j, Suppl. Fig. 1d). The slightly lower velocities observed in these mice may result from differences in behavior or technique. First, the mice with iAChSnFR were imaged only while immobile whereas those imaged with GRAB-ACh3.0 were allowed to run on a treadmill, which may be associated with faster waves. Second, because the former were imaged via a smaller (1 mm diameter) imaging aperture, the algorithm used to identify the waves may fail to identify waves whose spatial scale is larger than the aperture.

Reviewer comment

- Fig 3 is also a new addition. For 6 neurons (CINs), out of 29 neurons from a total of 13 mice, some correlation of activation of a single CIN leading to an increase in GRAB-DA fluorescence is observed. The data in the figure contain selected traces from 4 neurons/imaging fields. Each example uses a different stimulation protocol, it is unclear whether within an image area the phenomenon is repeatable, and no attempt for establishing causality (for example a before-after experiment with nAChR blockers) is made. The authors conclude, quite strongly, that "activation of a single CIN suffices to induces local striatal DA release" (lines 81-82) and "...overturning the widely-held view that synchronous activation of several CINs is required" (lines 431-432). While this would be a very important conclusion, it is not scientific to use the presented data to make this conclusion so strongly. As it stands, it is a correlation in a few example traces without an attempt to establish causality and reproducibility, and for most neurons it appears not to be the case. Systematic experimentation and analyses would be required.

Author response

We have performed additional systematic experiments and analyses to address this comment and now base our conclusions on analysis of 41 patched CINs from 18 mice. We find that in 24% of the patched CINs (10/41) a burst of action potentials in a single CIN caused a measurable increase in DA in the vicinity of the CIN. Statistical analysis ruled out the possibility that this was circumstantial coincidence of spontaneous release. In half of the CINs in which we were able to elicit a response (5/10) we were able to do it multiple times on the same cell, demonstrating that the phenomenon is reasonably reliable. We tested the effect of blocking nicotinic receptors with mecamylamine in one of these CINs and found that it blocked DA release (see new Fig. 3e). The nAChR-dependence of DA release was buttressed by additional experiments replicating previous findings showing that mecamylamine blocks DA release evoked by synchronous optogenetic activation of CINs (Suppl. Fig 2a). We also conducted new experiments to show that ACh release by a single CIN is much more reliable and exhibits no failures, demonstrating that the low rate of DA release evoked by individual CINs is attributable to the properties of DA release sites. We support this conclusion with existing literature concerning release of DA from individual terminals. We conclude that we have now clearly demonstrated that we did not observe a mere correlation between CIN activation and DA release, but rather show that individual CINs can cause DA release, oftentimes reliably, albeit infrequently. Moreover, we have revealed that it is DA axons that are responsible for the failures. We report these extended results as follows (lines 165-212, in the clean copy):

The prevailing view is that only synchronous activation of multiple CINs can induce localized DA release^{21, 22, 23, 26}. Indeed, we confirmed that synchronous optogenetic activation of CINs induced robust nAChR-dependent DA release (Suppl. Fig. 2a).

To test if activation of a single CIN can also induce DA release, we combined two-photon laser scanning microscopy (2PLSM) imaging of GRAB-DA2m, expressed selectively in dopaminergic axons, with patch-clamp recordings from individual CINs in acute striatal slices (Fig. 3a). We found that in 24% of the patched CINs ($n = 10/41$ CINs from $N = 10/18$ mice), evoking a burst of action potentials (APs) caused a measurable increase in DA in the vicinity of the CIN. Measurements in various regions-of-interest within the axonal arbor showed DA release throughout the CIN's arbor

(Fig. 3b), suggesting that a single CIN can influence DA release as far as its axonal field extends. In some cases, we were able to locate an individual stretch of DA axon that exhibits an even larger amplitude of release, presumably because the DA concentration is highest near the releasing axon that expressed the DA sensor (Fig. 3b, red region-of-interest). In 2 CINs, we were even able to observe DA release in response to a single AP (Fig. 3c). DA release could also be observed in 3 CINs in response to rebound spiking after hyperpolarizing the CIN, which may be important in the context of the pause response exhibited by CINs^{31, 41, 42}. In 5 of the 10 CINs whose stimulation evoked DA release, the release occurred on multiple trials (that had to be separated by > 1 min long intervals) within the nearby DA axonal arbor, although never at the same exact location. We confirmed in one of these CINs that DA release was blocked by the nAChR blocker mecamylamine (Fig. 3e), just as with the synchronous activation of CINs (Suppl. Fig. 2a).

To determine whether the frequent failures in DA release resulted from the unreliability of ACh release in response to activation of a single CIN, we repeated the above experiment but expressed GRAB-ACh3.0 in the dorsal striatum. We found that repeated activation of an individual CIN (every 5 seconds) reliably released ACh each time (Suppl. Fig. 2b). This suggests that the low repeatability of DA release that we observed is due to refractoriness of DA release sites after release, in line with previous studies. With repeated stimuli, DA release decreases sharply after the first stimulus^{28, 43, 44, 45} and stays decreased for up to 60 seconds⁴⁴. Fluorescent false neurotransmitter experiments indicate that a single stimulus causes exocytosis of a large fraction of releasable vesicles (17%)⁴⁶ leading to a sharp decrease in DA release in response to subsequent stimuli^{28, 43, 45}. Liu et al.⁴⁷ show that only the first action potential of a sequence triggers DA release. Moreover, the dopaminergic vesicle pool is slow to replenish (with a time constant of ~21 s)⁴⁸. Finally, desensitization of nAChRs, which occurs more readily in acute striatal slices^{28, 45, 49} may contribute as well.

Interestingly, we also observed spontaneous DA release events (e.g., Fig. 3e) that were not triggered within 100 ms of our stimulation, raising the possibility that the evoked responses were actually spontaneous ones that spuriously coincided with our stimulation. However, the observed rate of occurrence of DA release events is on average approximately 1 event per minute (i.e., a total of 87 spontaneous plus evoked events occurring during the cumulative 85.32 minutes of imaging, which). This rate is 30 times lower than reported in whole slice imaging²¹, but the area we imaged is typically two orders of magnitude smaller than the area imaged in that study, which can account for that discrepancy. With a rate of 1 event per minute, only 1.5 spontaneous events would be expected to occur within 100 ms of the 878 stimulation events delivered (i.e., 87 DA events × 878 stimulation events × 0.1 s coincidence window / 5119.2 s), ruling out that the 26 evoked responses were spontaneous ones.

Reviewer comment

At best, the current data suggest that rarely, a single CIN might suffice to induce dopamine release, but in most cases it might not. At this point, the data are circumstantial.

Author response

The reviewer's critique forced us to better substantiate our claim that single CINs can drive local DA release with new data and analyses (see above). Nevertheless, in light of their comment, we toned down the conclusion and now state in the Discussion (lines 467-9, in the clean copy):

Finally, we demonstrated in acute striatal slices for the first time that APs elicited in individual CINs can produce local DA release, albeit infrequently.

Reviewer comment

Model assumption: the model assumes that DA and ACh axonal arbors interact at all points. This is not the case. Direct connections are very sparse (PMID: 8684624), and the striatum has a high

concentration of acetylcholine esterase and inhibiting it affects both the frequency and kinetics of ACh-to-DA transmission (PMID: 35931070). Clearly, their interactions are limited.

Author response

We appreciate the reviewer's concern. We failed to explain clearly what we meant by "interact at all points". Here, we do not intend "points" to mean "direct synaptic connections". Rather, we mean "volumes sufficiently large to support an interaction". Our reading of the evidence suggests that these volumes are small and contiguous. To clarify our intent, we rewrote the Discussion as follows (lines 575-619):

We based the diffusion component on the density and intermingling of ACh and DA terminals, the affinity and proximity of DA terminal nicotinic receptors to ACh release sites, and the affinity and concentration of acetylcholinesterase (AChE), as detailed below. These considerations suggest that within a volume of a few μm^3 there are sufficient numbers of ACh and DA terminals in near enough proximity to ensure interactions between them, and such contiguous volumes ensure ACh-DA interactions can occur continuously throughout the striatum. The key facts are as follows: The density of choline acetyltransferase (ChAT)-positive varicosities containing ACh vesicles is $0.20 \mu\text{m}^{-3}$ of which 10% display synaptic membrane specializations⁸⁰, giving an estimate of the density of ACh release sites of $\rho_{\text{ACh}} = 0.020 \mu\text{m}^{-3}$. This is consistent with estimates⁵⁷ based on the quantitative electron microscopy measures of density of symmetric synapses⁸¹ and the fraction of symmetric synapses that are cholinergic⁸². If randomly arranged this density of release sites predicts an average nearest neighbor distance of $2.04 \mu\text{m}$ using the method of Clark and Evans^{83, 84}. The corresponding density of DA varicosities is $0.10 \mu\text{m}^{-3}$ ^{6, 85, 86}. Assuming 17% of varicosities are active release sites⁴⁶ the density of DA release sites is about $\rho_{\text{DA}} = 0.017 \mu\text{m}^{-3}$ and if randomly arranged the average nearest neighbor distance between DA release sites is $2.10 \mu\text{m}$, which is consistent with the spacing of DA synapses measured in serial sections by electron microscopy⁸⁷. Combining both ACh and DA release site densities we might expect a mean distance from ACh release sites to the nearest DA release site to be on the order of $1.6 \mu\text{m}$. This is consistent with recent direct, super-resolution measurements indicating the distance of ACh terminals from their nearest DA axon is less than $0.5 \mu\text{m}$ in half of the cases and less than $2.0 \mu\text{m}$ in almost all cases²¹. Thus, there is a high probability of a DA release site within a sphere of $2.0 \mu\text{m}$ radius around each ACh terminal.

The spread and time course of ACh depends on diffusion from release sites and on the activity and concentration of AChE. Reported values of K_m for AChE range from 0.5×10^{-4} to $1.4 \times 10^{-3} \text{M}$ ^{88, 89} which are much higher than the concentration of ACh and affinity of ACh receptors. Although AChE is one of the fastest enzymes known^{90, 91} the effective turnover number is low at low levels of ACh. Quite high concentrations of AChE (micromolar range) would be required to hydrolyse ACh in a millisecond timeframe⁹². While such high concentrations of AChE exist in the neuromuscular junction, in the brain major AChE form in the CNS is an amphiphilic globular tetramer ($G_4 \text{AChE}$)^{93, 94} that is present at much lower concentrations on the order of 300nM ⁹⁵. Calculations of the spatial spread and time course of ACh⁵⁷ indicate that striatal AChE would have small effects on immediate time course (ms range) of ACh concentration in the near distance (μm range) after a release event and its main effect would be on the ambient levels of ACh at longer delays (seconds) and distances (tens of μm). Moreover the majority (85%) of DA boutons contain the nicotinic ACh receptor $\beta 2$ subunit⁹⁶. The $\alpha 4\beta 2$ nicotinic receptors found *in vivo* exhibit high agonist sensitivity (EC_{50} for ACh of $0.97 \mu\text{M}$)⁹⁷. Thus, $\alpha 4\beta 2$ nicotinic ACh receptors would be functionally effective even if located several micrometers from release sites. Consistent with this, Kramer et al²⁶ found that blocking AChE did not potentiate the peak amplitude of stimulated or spontaneous ACh mediated EPSPs in DA axons but caused a later reduction in the peak amplitudes, likely due to receptor desensitization from the increased ambient concentration. They concluded that their results "suggest an arrangement whereby a fraction of CIN terminals exist in close proximity to nAChRs on dopaminergic axons".

We also clarified this point when defining the reaction diffusion system as follows (lines 233-5, in the clean copy):

This structure combined with the fact that DA and ACh may diffuse some distance from release sites, lends support to our modeling them as continuous media (or syncytia) that fill space and interact within small, contiguous volumes.

Reviewer comment

- The simplest and most straightforward prediction of the model is that inhibiting ACh receptors should impair dopamine waves. Without that test, much of the model remains unvalidated.

Author response

We agree that this is a simple and straightforward prediction of the model. However, testing this prediction experimentally is neither simple nor straightforward, as it would require a) establishing conditions for reliable *in vivo* imaging of DA waves in the awake animal configuration, b) determining appropriate local pharmacological approaches, doses, timing, and selectivity of drugs to inhibit nicotinic ACh receptors (nAChRs), and c) defining appropriate experiments to control for off-target drug effects (i.e. multiple agonist, antagonist to confirm drug effect is on nAChR), and for indirect effects of such inhibition (i.e. drug-induced changes in behavior or attention) that might also account for changes in dopamine waves. We agree that this would be a great project that we hope to be able to carry out in the future. At this point however, we feel it would be a major undertaking and, in many ways, beyond the scope of the present paper.

Reviewer comment

- Clear descriptions of the number of observations and statistics in the figure legends are missing.

Author response

The number our observations and statistics appear in the main text, in accordance with the Guide for Authors. We include n's and N's in the legends of supplementary figures (eg., suppl. Figs. 2a and 2b).

Reviewer comment

- In several figures (for example 1f-1j), y-axis labels are missing.

Author response

We thank the Reviewer for spotting this oversight, which we have now corrected.

Reviewer comment

Altogether, there remain significant weaknesses in the biological data and in the assessment of their relationship to the model. The existence of ACh waves and their influence on DA waves remains uncertain.

Author response

We appreciate the Referee's constructive critique and hope he/she will find that our study has been strengthened by the suggested revisions.

Reviewer #3

Reviewer comment

Understanding DA wave mechanisms is critical for the understanding of striatal function and dysfunction. This manuscript suggests a possible mechanism underlying this wave's existence. My suggestions were focused on alternative mechanisms of these waves. These mechanisms were discussed and modeled in the revised manuscript. I hope this work will inspire new experimental work to prove or improve the model. In my view, this work is ready to be presented to the public.

Author response

We appreciate the Reviewer's time in reviewing our study and his/her enthusiasm about its publication.

Reviewers' Comments:

Reviewer #2:

Remarks to the Author:

Final comments: The authors have again made a significant effort to address my concerns. Overall, I think that it is not productive to continue to go through further rounds of review. In my view, many of the data remain circumstantial observations rather than conclusive or causative, and I will unlikely be convinced by the approach taken by the authors. The main disagreement lies in the numbers of observations for a given finding. My view is that an experiment should be repeated the same way so data can be analyzed, pooled and assessed quantitatively. In contrast, Figs. 1+3 each contain experiments that are often done similarly but not identically. In each figure, the data cannot be pooled and analyzed meaningfully for a sufficient number of observations. They end up being "case reports" in Fig. 3 (no repeats across cells) or too few repeats in Fig. 1 for each type of experiment (2 different sensors, two different imaging modalities, combined in various ways). I also acknowledge that the other two reviewers are in support of publishing this paper. Ultimately, the work should be published so that a broad readership can assess it and the authors and the field can continue to evaluate the interesting phenomena that are described and tested here. In that sense, I am fine with remaining in disagreement on the points raised throughout the review process and to have the paper published in Nature Communications.

Reviewer 2:

Final comments: The authors have again made a significant effort to address my concerns. Overall, I think that it is not productive to continue to go through further rounds of review. In my view, many of the data remain circumstantial observations rather than conclusive or causative, and I will unlikely be convinced by the approach taken by the authors. The main disagreement lies in the numbers of observations for a given finding. My view is that an experiment should be repeated the same way so data can be analyzed, pooled and assessed quantitatively. In contrast, Figs. 1+3 each contain experiments that are often done similarly but not identically. In each figure, the data cannot be pooled and analyzed meaningfully for a sufficient number of observations. They end up being “case reports” in Fig. 3 (no repeats across cells) or too few repeats in Fig. 1 for each type of experiment (2 different sensors, two different imaging modalities, combined in various ways). I also acknowledge that the other two reviewers are in support of publishing this paper. Ultimately, the work should be published so that a broad readership can assess it and the authors and the field can continue to evaluate the interesting phenomena that are described and tested here. In that sense, I am fine with remaining in disagreement on the points raised throughout the review process and to have the paper published in Nature Communications.

Response

We thank the Reviewer for the time and effort devoted to re-evaluating our revised study, and for graciously allowing it to move forward and be evaluated by the broader neuroscience community despite lingering methodological reservations. We agree that repeating experiment the same way so that data can be pooled is best. However, there is also strength in reporting similar findings obtained in two different labs using different sensors because it shows reproducibility as well as statistical significance. In our case, we feel that the data clearly show that wave-like behavior of striatal acetylcholine activity is a robust phenomenon that is reproduced across animals and experiments.

We also agree that it is good to show repeats across cells, and we feel we have done this. In Fig 3 we show repeats with the same exact stimulus – driving a burst of spiking – in all of the cholinergic interneurons recorded and we report pooled dopamine responses elicited from 10 neurons in 10 mice. We explain that a peculiarity of this experiment is that once dopamine is released in one region in the slice, it is difficult to elicit dopamine release in the same spot again. This long refractoriness of dopamine release has been reported by other labs as well, and we show that it contrasts with the release of acetylcholine, which can be repeated at short intervals. When querying dopamine release from several locations in the slice, we report being able to detect dopamine release multiple times from stimulating the same cell in 5 out of 10 recordings. Thus, in our opinion, given the physiological nature of this preparation, we have seen repeatability both between and within cells to a satisfactory extent.